# ComaDICE: Offline Cooperative Multi-Agent Reinforcement Learning with Stationary Distribution Shift Regularization

**The Viet Bui**
School of Computing and Information Systems
Singapore Management University, Singapore
`theviet.bui.2023@phdcs.smu.edu.sg`

**Hong Thanh Nguyen**
University of Oregon Eugene, Oregon
United States
`thanhhng@cs.orgeon.edu`

**Tien Mai**
School of Computing and Information Systems
Singapore Management University, Singapore
`atmai@smu.edu.sg`

## Abstract

Offline reinforcement learning (RL) has garnered significant attention for its ability to learn effective policies from pre-collected datasets without the need for further environmental interactions. While promising results have been demonstrated in single-agent settings, offline multi-agent reinforcement learning (MARL) presents additional challenges due to the large joint state-action space and the complexity of multi-agent behaviors. A key issue in offline RL is the *distributional shift*, which arises when the target policy being optimized deviates from the behavior policy that generated the data. This problem is exacerbated in MARL due to the interdependence between agents' local policies and the expansive joint state-action space. Prior approaches have primarily addressed this challenge by incorporating regularization in the space of either Q-functions or policies. In this work, we introduce a regularizer in the space of stationary distributions to better handle distributional shift. Our algorithm, ComaDICE, offers a principled framework for offline cooperative MARL by incorporating stationary distribution regularization for the global learning policy, complemented by a carefully structured multi-agent value decomposition strategy to facilitate multi-agent training. Through extensive experiments on the multi-agent *MuJoCo* and *StarCraft II* benchmarks, we demonstrate that ComaDICE achieves superior performance compared to state-of-the-art offline MARL methods across nearly all tasks.

## 1 Introduction

Over the years, deep RL has achieved remarkable success in various decision-making tasks (Levine et al., 2016; Silver et al., 2017; Kalashnikov et al., 2018; Haydari & Yılmaz, 2020). However, a significant limitation of deep RL is its need for millions of interactions with the environment to gather experiences for policy improvement. This process can be both costly and risky, especially in real-world applications like robotics and healthcare. To address this challenge, offline RL has emerged, enabling policy learning based solely on pre-collected demonstrations (Levine et al., 2020). Despite this advancement, offline RL faces a critical issue: the distribution shift between the offline dataset and the learned policy (Kumar et al., 2019). This distribution shift complicates value estimation for unseen states and actions during policy evaluation, resulting in extrapolation errors where out-of-distribution (OOD) state-action pairs are assigned unrealistic values (Fujimoto et al., 2018).

To tackle OOD actions, many existing works impose action-level constraints, either implicitly by regulating the learned value functions or explicitly through distance or divergence penalties (Fujimoto et al., 2019; Kumar et al., 2019; Wu et al., 2019; Peng et al., 2019; Fujimoto & Gu, 2021; Xu et al., 2021). Only a few recent studies have addressed both OOD actions and states using state-action-level

behavior constraints (Li et al., 2022; Zhang et al., 2022; Lee et al., 2021; 2022; Mao et al., 2024). In particular, there is an important line of work on DIstribution Correction Estimation (DICE) (Nachum & Dai, 2020) that constrains the distance in terms of the joint state-action occupancy measure between the learning policy and the offline policy. These DICE-based methods have demonstrated impressive performance results on the D4RL benchmarks (Lee et al., 2021; 2022; Mao et al., 2024).

It is important to note that that all the aforementioned offline RL approaches primarily focus on the single-agent setting. While multi-agent setting is prevalent in many real-world sequential decision-making tasks, offline MARL remains a relatively under-explored area. The multi-agent setting poses significantly greater challenges due to the large joint state-action space, which expands exponentially with the number of agents, as well as the inter-dependencies among the local policies of different agents. As a result, the offline data distribution can become quite sparse in these high-dimensional joint action spaces, leading to an increased number of OOD state-action pairs and exacerbating extrapolation errors. A few recent studies have sought to address the negative effects of sparse data distribution in offline MARL by adapting the well-known centralized training decentralized execution (CTDE) paradigm from online MARL (Oliehoek et al., 2008; Kraemer & Banerjee, 2016), enabling data-related regularization at the individual agent level. Notably, some of these works (Pan et al., 2022; Shao et al., 2024; Wang et al., 2022b) extend popular offline single-agent RL algorithms, such as CQL (Kumar et al., 2020) and SQL/EQL (Xu et al., 2023), within the CTDE framework.

In our work, we focus on addressing the aforementioned challenges in offline cooperative MARL. In particular, we follow the DICE approach to address both OOD states and actions, motivated by remarkable performance of recent DICE-based methods in offline single-agent RL. Similar to previous works in offline MARL, we adopt the CTDE framework to handle exponential joint state-action spaces in the multi-agent setting. We remark that extending the DICE approach under this CTDE framework is not straightforward given the complex objective of DICE that involves the f-divergence in stationary distribution between the learning joint policy and the behavior policy. Therefore, the value decomposition in CTDE needs to be carefully designed to ensure the consistency in optimality between the global and local policies. In particular, we provide the following main contributions:

- We propose ComaDICE, a new offline MARL algorithm that integrates DICE with a carefully designed value decomposition strategy. In ComaDICE, under the CTDE framework, we decompose both the global value function $\nu^{tot}$ and the global advantage functions $A_\nu^{tot}$, rather than using Q-functions as in previous MARL works. This unique factorization approach allows us to theoretically demonstrate that the global learning objective in DICE is convex in local values, provided that the mixing network used in the value decomposition employs non-negative weights and convex activation functions. This significant finding ensures that our decomposition strategy promotes an efficient and stable training process.

- Building on our decomposition strategy, we demonstrate that finding an optimal global policy can be divided into multiple sub-problems, each aims to identify a local optimal policy for an individual agent. We provide a theoretical proof that the global optimal policy is, in fact, equivalent to the product of the local policies derived from these sub-problems.

- Finally, we conduct extensive experiments to evaluate the performance of our algorithm, ComaDICE, in complex MARL environments, including: multi-agent StarCraft II (i.e., SMACv1 (Samvelyan et al., 2019), SMACv2 (Ellis et al., 2022)) and multi-agent Mujoco (de Witt et al., 2020) benchmarks. Our empirical results show that our ComaDICE outperforms several strong baselines in all these benchmarks.

## 2    RELATED WORK

**Offline Reinforcement Learning (offline RL).**    Offline RL focuses on learning policies from pre-collected datasets without any further interactions with the environment (Levine et al., 2020; Prudencio et al., 2023). A significant challenge in offline RL is the issue of distribution shift, where unseen actions and states may arise during training and execution, leading to inaccurate policy evaluations and suboptimal outcomes. Consequently, there is a substantial body of literature addressing this challenge through various approaches (Prudencio et al., 2023). In particular, some studies impose explicit or implicit policy constraints to ensure that the learned policy remains close to the behavioral policy (Fujimoto et al., 2019; Kumar et al., 2019; Wu et al., 2019; Kostrikov et al., 2021; Peng et al., 2019; Nair et al., 2020; Fujimoto & Gu, 2021; Xu et al., 2021; Cheng et al., 2024; Li

et al., 2023). Others incorporate regularization terms into the learning objectives to mitigate the value overestimation on OOD actions (Kumar et al., 2020; Kostrikov et al., 2021; Xu et al., 2022c; Niu et al., 2022; Xu et al., 2023; Wang et al., 2022b). Uncertainty-based offline RL methods seek to balance conservative approaches with naive off-policy RL techniques, relying on estimates of model, value, or policy uncertainty (Agarwal et al., 2020; An et al., 2021; Bai et al., 2022). Offline model-based algorithms focus on conservatively estimating the transition dynamics and reward functions based on the pre-collected datasets (Kidambi et al., 2020; Yu et al., 2020; Matsushima et al., 2020; Yu et al., 2021). Some other methods impose action-level regularization through imitation learning techniques (Xu et al., 2022b; Chen et al., 2020; Zhang et al., 2023; Zheng et al., 2024; Brandfonbrener et al., 2021; Xu et al., 2022a). Finally, while a majority of previous works target OOD actions only, there are a few recent works attempt to address both OOD states and actions (Li et al., 2022; Zhang et al., 2022; Lee et al., 2021; 2022; Sikchi et al., 2023; Mao et al., 2024). Our work on offline MARL follow the DICE-based approach, as motivated by compelling performance of DICE-based algorithms in single-agent settings (Lee et al., 2021; 2022; Sikchi et al., 2023; Mao et al., 2024).

**Offline Multi-agent Reinforcement Learning (offline MARL).** While there is a substantial body of literature on offline single-agent RL, research on offline MARL remains limited. Offline MARL faces challenges from both distribution shift—characteristic of offline settings—and the exponentially large joint action space typical of multi-agent environments. Recent studies have begun to merge advanced methodologies from both offline RL and MARL to address these challenges (Yang et al., 2021; Pan et al., 2022; Shao et al., 2024; Wang et al., 2022b) Specifically, these works employ local policy regularization within the centralized training with decentralized execution (CTDE) framework to mitigate distribution shift. The CTDE paradigm, well-established in online MARL, facilitates more efficient and stable learning while allowing agents to operate in a decentralized manner (Oliehoek et al., 2008; Kraemer & Banerjee, 2016). For instance, Yang et al. (2021) utilize importance sampling to manage local policy learning on OOD samples. Both works by Pan et al. (2022) and Shao et al. (2024) are built upon CQL (Kumar et al., 2020), a prominent offline RL algorithm for single-agent scenarios. Matsunaga et al. (2023) developed AlberDICE, leveraging the Nash equilibrium solution concept from game theory to iteratively update the best responses of individual agents. Both AlberDICE and our method, ComaDICE, adopt the DICE framework to address the out-of-distribution (OOD) issue. However, while AlberDICE proposes learning individual Lagrange multipliers (or value functions) to obtain occupancy ratios, our ComaDICE algorithm learns a global value function by mixing local functions, adhering to the well-established CTDE principle. This design enables ComaDICE to better capture inter-agent relationships and improve credit assignment across local agents. Finally, OMIGA (Wang et al., 2022b) establishes the equivalence between global and local value regularization within a *policy constraint framework*, making it the current state-of-the-art algorithm in offline MARL. The key difference between ComaDICE and OMIGA lies in their respective approaches: OMIGA focuses on learning a global Q-function, whereas our algorithm (and other methods in the DICE family) operates in the occupancy space, aiming to learn the ratio between the occupancy of the learning policy and the behavior policy.

Beyond this main line of research, some studies formulate offline MARL as a sequence modeling problem, employing supervised learning techniques to tackle the issue (Meng et al., 2023; Tseng et al., 2022), while others adhere to decentralized approaches (Jiang & Lu, 2023).

## 3 PRELIMINARIES

Our work focuses on cooperative multi-agent RL, which can be modeled as a multi-agent Partially Observable Markov Decision Process (POMDP), defined by the tuple $\mathcal{M} = \langle \mathcal{S}, \mathcal{A}, P, r, \mathcal{Z}, \mathcal{O}, n, \mathcal{N}, \gamma \rangle$. Here, $n$ is number of agents, $\mathcal{N} = \{1, \ldots, n\}$ is the set of agents, $\mathbf{s} \in S$ represents the true state of the multi-agent environment, and $\mathcal{A} = \prod_{i \in \mathcal{N}} \mathcal{A}_i$ is the set of joint actions, where $\mathcal{A}_i$ is the set of individual actions available to agent $i \in \mathcal{N}$. At each time step, each agent $i \in \{1, 2, \ldots, n\}$ selects an action $a_i \in \mathcal{A}_i$, forming a joint action $\mathbf{a} = (a_1, a_2, \ldots, a_n) \in \mathcal{A}$. The transition dynamics $P(\mathbf{s}'|\mathbf{s}, \mathbf{a}) : \mathcal{S} \times \mathcal{A} \times \mathcal{S} \to [0, 1]$ describe the probability of transitioning to the next state $\mathbf{s}'$ when agents take an action $\mathbf{a}$ from the current state $\mathbf{s}$. The discount factor $\gamma \in [0, 1)$ represents the weight given to future rewards. In a partially observable environment, each agent receives a local observation $s_i \in \mathcal{O}_i$ based on the observation function $\mathcal{Z}_i(\mathbf{s}) : \mathcal{S} \to \mathcal{O}_i$, and we denote the joint observation as $\mathbf{o} = (o_1, o_2, \ldots, o_n)$. In cooperative MARL, all agents share a global reward

function $r(\mathbf{s}, \mathbf{a}) : \mathcal{S} \times \mathcal{A} \rightarrow \mathbb{R}$. The goal of all agents is to learn a joint policy $\boldsymbol{\pi}_{\text{tot}} = \{\pi_1, \dots, \pi_n\}$ that collectively maximize the expected discounted returns $\mathbb{E}_{(\mathbf{o}, \mathbf{a}) \sim \boldsymbol{\pi}_{\text{tot}}} [\sum_{t=0}^{\infty} \gamma^t r(\mathbf{s}_t, \mathbf{a}_t)]$. In the offline MARL setting, a pre-collected dataset $\mathcal{D}$ is obtained by sampling from a behavior policy $\boldsymbol{\mu}_{\text{tot}} = \{\mu_1, \dots, \mu_n\}$, and the policy learning is conducted soly based on $\mathcal{D}$, with no interactions with the environment. We also define the occupancy measure (or stationary distribution) as follows:

$$\rho^{\boldsymbol{\pi}_{tot}}(\mathbf{s}, \mathbf{a}) = (1 - \gamma) \sum\nolimits_{t=0}^{\infty} P(\mathbf{s}_t = \mathbf{s}, \ \mathbf{a}_t = \mathbf{a})$$

which represents distribution visiting the pair (observation, action) $(\mathbf{s}_t, \mathbf{a}_1)$ when following the joint policy $\boldsymbol{\pi}_{tot}$, where $\mathbf{s}_0 \sim P_0$, $\mathbf{a}_t \sim \boldsymbol{\pi}_{tot}(\cdot | \mathbf{s}_t)$ and $\mathbf{s}_{t+1} \sim P(\cdot | \mathbf{s}_t, \mathbf{a}_t)$.

# 4 COMADICE: OFFLINE COOPERATIVE MULTI-AGENT RL WITH STATIONARY DISTRIBUTION CORRECTION ESTIMATION

We consider an offline cooperative MARL problem where the goal is to optimize the expected discounted joint reward. In this work, we focus on the DICE objective function Nachum & Dai (2020); Lee et al. (2021), which incorporates a stationary distribution regularizer to capture the divergence between the occupancy measures of the learning policy, $\boldsymbol{\pi}_{tot}$, and the behavior policy, $\boldsymbol{\mu}_{tot}$, formulated as follows:

$$\max_{\boldsymbol{\pi}_{tot}} \quad \mathbb{E}_{(\mathbf{s}, \mathbf{a}) \sim \rho^{\boldsymbol{\pi}_{tot}}} [r(\mathbf{s}, \mathbf{a})] - \alpha D^f \left( \rho^{\boldsymbol{\pi}_{tot}} \| \rho^{\boldsymbol{\mu}_{tot}} \right) \tag{1}$$

where $D^f \left( \rho^{\boldsymbol{\pi}_{tot}} \| \rho^{\boldsymbol{\mu}_{tot}} \right) = \mathbb{E}_{(\mathbf{s}, \mathbf{a}) \sim \rho^{\boldsymbol{\pi}_{tot}}} \left[ f \left( \frac{\rho^{\boldsymbol{\pi}_{tot}}}{\rho^{\boldsymbol{\mu}_{tot}}} \right) \right]$ is the f-divergence between the stationary distribution $\rho^{\boldsymbol{\pi}_{tot}}$ of the learning policy and $\rho^{\boldsymbol{\mu}_{tot}}$ of the behavior policy. In this work, we consider $f(\cdot)$ to be strictly convex and differentiable. The parameter $\alpha$ controls the trade-off between maximizing the reward and penalizing deviation from the offline dataset's distribution (i.e., penalizing distributional shift). When $\alpha = 0$, the problem becomes the standard offline MARL, where the objective is to find a joint policy that maximizes the expected joint reward. On the other hand, when $\alpha \gg 1$, the problem shifts towards imitation learning, aiming to closely mimic the behavioral policy.

This DICE-based approach offers the advantage of better capturing the system dynamics inherent in the offline data. Such stationary distributions, $\rho^{\boldsymbol{\pi}_{tot}}$ and $\rho^{\boldsymbol{\mu}_{tot}}$, however, are not directly available. We will discuss how to estimate them in the next subsection.

## 4.1 CONSTRAINED OPTIMIZATION IN THE STATIONARY DISTRIBUTION SPACE

We first formulate the learning problem in Eq. 1 as a constrained optimization on the space of $\rho^{\boldsymbol{\pi}_{tot}}$:

$$\max_{\rho^{\boldsymbol{\pi}_{tot}}} \quad \mathbb{E}_{(\mathbf{s}, \mathbf{a}) \sim \rho^{\boldsymbol{\pi}_{tot}}} [r(\mathbf{s}, \mathbf{a})] - \alpha D^f \left( \rho^{\boldsymbol{\pi}_{tot}} \| \rho^{\boldsymbol{\mu}_{tot}} \right) \tag{2}$$

$$s.t. \quad \sum\nolimits_{\mathbf{a}'} \rho^{\boldsymbol{\pi}_{tot}}(\mathbf{s}, \mathbf{a}') = (1 - \gamma) p_0(\mathbf{s}) + \gamma \sum\nolimits_{\mathbf{a}', \mathbf{s}'} \rho^{\boldsymbol{\pi}_{tot}}(\mathbf{s}', \mathbf{a}') P(\mathbf{s} | \mathbf{a}', \mathbf{s}'), \ \forall \mathbf{s} \in \mathcal{S}. \tag{3}$$

When $f$ is convex, (2-3) becomes a convex optimization problem, as it involves maximizing a concave objective function subject to linear constraints. We now consider the Lagrange dual of (2-3):

$$\mathcal{L}(\nu^{tot}, \rho^{\boldsymbol{\pi}_{tot}}) = \mathbb{E}_{(\mathbf{s}, \mathbf{a}) \sim \rho^{\boldsymbol{\pi}_{tot}}} [r(\mathbf{s}, \mathbf{a})] - \alpha \mathbb{E}_{(\mathbf{s}, \mathbf{a}) \sim \rho^{\boldsymbol{\mu}_{tot}}} \left[ f \left( \frac{\rho^{\boldsymbol{\pi}_{tot}}(\mathbf{s}, \mathbf{a})}{\rho^{\boldsymbol{\mu}_{tot}}(\mathbf{s}, \mathbf{a})} \right) \right]$$

$$- \sum_{\mathbf{s}} \nu^{tot}(\mathbf{s}) \Big( \sum\nolimits_{\mathbf{a}'} \rho^{\boldsymbol{\pi}_{tot}}(\mathbf{s}, \mathbf{a}') - (1 - \gamma) p_0(\mathbf{s}) - \gamma \sum\nolimits_{\mathbf{a}', \mathbf{s}'} \rho^{\boldsymbol{\pi}_{tot}}(\mathbf{s}', \mathbf{a}') P(\mathbf{s} | \mathbf{a}', \mathbf{s}') \Big), \tag{4}$$

where $\nu^{tot}(\mathbf{s})$ is a Lagrange multiplier. Since (2-3) is a convex optimization problem, it is equivalent to the following minimax problem over the spaces of $\nu^{tot}$ and $\rho^{\boldsymbol{\pi}_{tot}}$: $\min_{\nu^{tot}} \max_{\rho^{\boldsymbol{\pi}_{tot}}} \{ \mathcal{L}(\nu^{tot}, \rho^{\boldsymbol{\pi}_{tot}}) \}$. Furthermore, we observe that $\mathcal{L}(\nu^{tot}, \rho^{\boldsymbol{\pi}_{tot}})$ is linear in $\nu^{tot}$ and concave in $\rho^{\boldsymbol{\pi}_{tot}}$, so the minimax problem has a saddle point, implying: $\min_{\nu^{tot}} \max_{\rho^{\boldsymbol{\pi}_{tot}}} \{ \mathcal{L}(\nu^{tot}, \rho^{\boldsymbol{\pi}_{tot}}) \} = \max_{\rho^{\boldsymbol{\pi}_{tot}}} \min_{\nu^{tot}} \{ \mathcal{L}(\nu^{tot}, \rho^{\boldsymbol{\pi}_{tot}}) \}$. In a manner analogous to the single-agent case (Lee et al., 2021), by defining $w_\nu^{tot}(\mathbf{s}, \mathbf{a}) = \frac{\rho^{\boldsymbol{\pi}_{tot}}(\mathbf{s}, \mathbf{a})}{\rho^{\boldsymbol{\mu}_{tot}}(\mathbf{s}, \mathbf{a})}$, the Lagrange dual function can be simplified into the more compact form (with detailed derivations are in the appendix):

$$\mathcal{L}(\nu^{tot}, w^{tot}) = (1 - \gamma) \mathbb{E}_{\mathbf{s} \sim p_0} [\nu^{tot}(\mathbf{s})] + \mathbb{E}_{(\mathbf{s}, \mathbf{a}) \sim \rho^{\boldsymbol{\mu}_{tot}}} \left[ -\alpha f \left( w_\nu^{tot}(\mathbf{s}, \mathbf{a}) \right) + w_\nu^{tot}(\mathbf{s}, \mathbf{a}) A_\nu^{tot}(\mathbf{s}, \mathbf{a}) \right],$$

where $A_\nu^{tot}$ is an "advantage function" defined based on $\nu^{tot}$ as:

$$A_\nu^{tot}(\mathbf{s}, \mathbf{a}) = q^{tot}(\mathbf{s}, \mathbf{a}) - \nu^{tot}(\mathbf{s}), \tag{5}$$

with $q^{tot}(\mathbf{s}, \mathbf{a}) = r(\mathcal{Z}(\mathbf{s}), \mathbf{a}) + \gamma \mathbb{E}_{\mathbf{s}' \sim P(\cdot|\mathbf{s}, \mathbf{a})}[\nu^{tot}(\mathbf{s}')]$. It is important to note that $\nu^{tot}(\mathbf{s})$ and $q^{tot}(\mathbf{s}, \mathbf{a})$ can be interpreted as a value function and a Q function, respectively, arising from the decomposition of the stationary distribution regularizer. We can now write the learning problem as follows:

$$\min_{\nu^{tot}} \max_{w^{tot} \geq 0} \; \{\mathcal{L}(\nu^{tot}, w^{tot})\}. \tag{6}$$

It can be observed that $\mathcal{L}(\nu^{tot}, w^{tot})$ is linear in $\nu^{tot}$ and concave in $w^{tot}$, which ensures well-behaved properties in both the $\nu^{tot}$- and $w^{tot}$-spaces. Following the derivations in Lee et al. (2021), a key feature of the above minimax problem is that the inner maximization problem has a closed-form solution, which greatly simplifies the minimax problem, making it no longer adversarial. We formalize this result as follows:

**Proposition 4.1.** *The minimax problem in Eq. 6 is equivalent to* $\min_{\nu^{tot}} \; \{\widetilde{\mathcal{L}}(\nu^{tot})\}$, *where*

$$\widetilde{\mathcal{L}}(\nu^{tot}) = (1 - \gamma)\mathbb{E}_{\mathbf{s} \sim p_0}[\nu^{tot}(\mathbf{s})] + \mathbb{E}_{(\mathbf{s}, \mathbf{a}) \sim \rho^{\boldsymbol{\mu}_{tot}}}\left[\alpha f^*\left(\frac{A_\nu^{tot}(\mathbf{s}, \mathbf{a})}{\alpha}\right)\right].$$

*Here, $f^*$ is convex conjugate of $f$, i.e., $f^*(y) = \sup_{t \geq 0}\{ty - f(t)\}$. Moreover, if $\nu^{tot}$ is parameterized by $\theta$, the first-order derivative of $\widetilde{\mathcal{L}}(\nu^{tot})$ w.r.t. $\theta$ is given as follows:*

$$\nabla_\theta \widetilde{\mathcal{L}}(\nu^{tot}) = (1 - \gamma)\mathbb{E}_{\mathbf{s} \sim p_0}[\nabla_\theta \nu^{tot}(\mathbf{s})] + \mathbb{E}_{(\mathbf{s}, \mathbf{a}) \sim \rho^{\boldsymbol{\mu}_{tot}}}\left[\nabla_\theta A_\nu^{tot}(\mathbf{s}, \mathbf{a})w_\nu^{tot*}(\mathbf{s}, \mathbf{a})\right].$$

*where $w_\nu^{tot*}(s, a) = \max\{0, f'^{-1}(A_\nu^{tot}(\mathbf{s}, \mathbf{a})/\alpha)\}$, with $f'^{-1}(\cdot)$ is the inverse function of the first-order derivative of $f$.*

Proposition 4.1 above is a direct extension of the formulations in Lee et al. (2021) developed for the single-agent setting, differing only in the inclusion of the closed-form expression for the first-order derivative of the objective function, $\widetilde{\mathcal{L}}(\nu^{tot})$.

## 4.2 Value Factorization

Directly optimizing $\min_{\nu^{tot}} \; \{\mathcal{L}(\nu^{tot}, w_\nu^{tot*})\}$ in multi-agent settings is generally impractical due to the large state and action spaces. Therefore, we follow the idea of value decomposition in the well-known CTDE framework in cooperative MARL to address this computational challenge. However, it is not straightforward to extend the DICE approach within this CTDE framework due to the complex objective of DICE, which involves the f-divergence between the learned joint policy and the behavior policy in stationary distributions. Thus, it is crucial to carefully design the value decomposition in CTDE to ensure optimality consistency between the global and local policies.

Specifically, we adopt a factorization approach that decomposes the value function $\nu^{tot}(\mathbf{s})$ (or global Lagrange multipliers) into local values using mixing network architectures. Let $\boldsymbol{\nu}(\mathbf{s}) = \{\nu_1(s_1), \ldots, \nu_n(s_n)\}$ represent a collection of local "value functions" and let $\mathbf{A}_{\boldsymbol{\nu}}(\mathbf{s}, \mathbf{a}) = \{A_i(s_i, a_i), \; i = 1, ..., n\}$ represent a collection of local advantage functions. The local advantage functions are computed as $A_i(s_i, a_i) = q_i(s_i, a_i) - \nu_i(s_i)$ for all $i \in \mathcal{N}$, where $\mathbf{q}(\mathbf{s}, \mathbf{a}) = \{q_i(s_i, a_i), \; i = 1, ..., n\}$ is a vector of local Q functions. To facilitate centralized learning, we create a mixing network, $\mathcal{M}_\theta$, where $\theta$ are the learnable weights, that aggregates the local values to form the global value and advantage functions as follows:

$$\nu^{tot}(\mathbf{s}, \mathbf{a}) = \mathcal{M}_\theta[\boldsymbol{\nu}(\mathbf{s})], \quad A_\nu^{tot}(\mathbf{s}, \mathbf{a}) = \mathcal{M}_\theta[\mathbf{q}(\mathbf{s}, \mathbf{a}) - \boldsymbol{\nu}(\mathbf{s})],$$

where each network takes the vectors $\boldsymbol{\nu}(\mathbf{s})$ or $\mathbf{A}_{\boldsymbol{\nu}}(\mathbf{s}, \mathbf{a})$ as inputs and outputs $\nu^{tot}$ and $A_\nu^{tot}$, respectively. Under this architecture, the learning objective becomes:

$$\widetilde{\mathcal{L}}(\boldsymbol{\nu}, \theta) = (1 - \gamma)\mathbb{E}_{\mathbf{s} \sim p_0}[\mathcal{M}_\theta[\boldsymbol{\nu}(\mathbf{s})]] + \mathbb{E}_{(\mathbf{s}, \mathbf{a}) \sim \rho^{\boldsymbol{\mu}_{tot}}}\left[\alpha f^*\left(\frac{\mathcal{M}_\theta[\mathbf{q}(\mathbf{s}, \mathbf{a}) - \boldsymbol{\nu}(\mathbf{s})]}{\alpha}\right)\right],$$

with the observation that $\mathbf{A}_\nu(\mathbf{s}, \mathbf{a})$ can be expressed as a linear function of $\boldsymbol{\nu}$. There are different ways to construct the mixing network $\mathcal{M}_\theta$; previous work often employs a single linear combination (1-layer network) or a two-layer network with convex activations such as ReLU, ELU, or Maxout. In the following, we show a general result stating that the learning objective function is convex in $\boldsymbol{\nu}$, provided that the mixing network is constructed with nonnegative weights and convex activations.

**Theorem 4.2.** *If the mixing network $\mathcal{M}_\theta[\cdot]$ is constructed with non-negative weights and convex activations, then $\widetilde{\mathcal{L}}(\boldsymbol{\nu}, \theta)$ is convex in $\boldsymbol{\nu}$.*

Mixing networks with non-negative weights and concave activations (e.g., ELU or ReLU) have been extensively used in MARL, forming the foundation of several notable state-of-the-art algorithms such as QMIX (Rashid et al., 2020), QTRAN (Son et al., 2019), and MFIQ (Bui et al., 2024). In particular, it has been demonstrated that mixing networks with either negative weights or non-concave activations result in significantly degraded performance (Bui et al., 2024). Theorem 4.2 shows that $\widetilde{\mathcal{L}}(\boldsymbol{\nu}, \theta)$ is convex in $\boldsymbol{\nu}$ when using *any multi-layer feed-forward mixing networks with non-negative weights and convex activation functions*. This finding is highly general and non-trivial, given the nonlinearity and complexity of both the function (in terms of $\boldsymbol{\nu}$) and the mixing networks. Previous work has often focused on single-layer (Wang et al., 2022b) or two-layer mixing structures (Rashid et al., 2020; Bui et al., 2024), emphasizing that such two-layer networks can approximate any monotonic function arbitrarily closely as network width approaches infinity (Dugas et al., 2009). In our experiments, we test two configurations for the mixing network: a linear combination (or 1-layer) and a 2-layer feed-forward network. While 2-layer mixing structures have shown strong performance in online MARL (Rashid et al., 2020; Son et al., 2019; Wang et al., 2020), we observe in our offline settings that the linear combination approach provides more stable results.

### 4.3 POLICY EXTRACTION

Let $\boldsymbol{\nu}^*$ be an optimal solution to the training problem with mixing networks, i.e.,

$$\min_{\boldsymbol{\nu}, \theta} \widetilde{\mathcal{L}}(\boldsymbol{\nu}, \theta). \tag{7}$$

We now need to extract a local and joint policy from this solution. Based on Prop. 4.1, given $\boldsymbol{\nu}^*$, we can compute this occupancy ratio as follows: : $w^{tot*}(\mathbf{s}, \mathbf{a}) = \max\left\{0, f'^{-1}\left(\frac{\mathcal{M}_\theta[\mathbf{A}_{\boldsymbol{\nu}^*}(\mathbf{s}, \mathbf{a})]}{\alpha}\right)\right\}$. The global policy can then be obtained as follows: $\boldsymbol{\pi}_{tot}^*(\mathbf{a}|\mathbf{s}) = \frac{w^{tot*}(\mathbf{s}, \mathbf{a}) \cdot \rho^{\boldsymbol{\mu}_{tot}}(\mathbf{s}, \mathbf{a})}{\sum_{\mathbf{a}' \in \mathcal{A}} w^{tot*}(\mathbf{s}, \mathbf{a}') \cdot \rho^{\boldsymbol{\mu}_{tot}}(\mathbf{s}, \mathbf{a}')}$. This computation, however, is not practical since $\rho^{\boldsymbol{\mu}_{tot}}$ is generally not available and might not be accurately estimated in the offline setting. A more practical way to estimate the global policy, $\boldsymbol{\pi}_{tot}^*$, as the result of solving the following weighted behavioral cloning (BC):

$$\max_{\boldsymbol{\pi}_{tot} \in \Pi_{tot}} \mathbb{E}_{(\mathbf{s}, \mathbf{a}) \sim \rho^{\boldsymbol{\pi}_{tot}^*}}[\log \boldsymbol{\pi}_{tot}(\mathbf{a}|\mathbf{s})] = \max_{\boldsymbol{\pi}_{tot} \in \Pi_{tot}} \mathbb{E}_{(\mathbf{s}, \mathbf{a}) \sim \rho^{\boldsymbol{\mu}_{tot}}}[w^{tot*}(\mathbf{s}, \mathbf{a}) \log \boldsymbol{\pi}_{tot}(\mathbf{a}|\mathbf{s})], \tag{8}$$

where $\Pi_{tot}$ represents the feasible set of global policies. Here we assume that $\Pi_{tot}$ contains decomposable global policies, i.e., $\Pi_{tot} = \{\boldsymbol{\pi}_{tot} \mid \exists \pi_i, \forall i \in \mathcal{N} \text{ such that } \boldsymbol{\pi}_{tot}(\mathbf{a}|\mathbf{s}) = \prod_{i \in \mathcal{N}} \pi_i(a_i|s_i)\}$. In other words, $\Pi_{tot}$ consists of global policies that can be expressed as a product of local policies. This decomposability is highly useful for decentralized learning and has been widely adopted in MARL (Wang et al., 2022b; Bui et al., 2024; Zhang et al., 2021).

While the above weighted BC appears practical, as $(\mathbf{s}, \mathbf{a})$ can be sampled from the offline dataset generated by $\rho^{\boldsymbol{\pi}_{tot}}$, and since $w^{tot*}(\mathbf{s}, \mathbf{a})$ is available from solving 7, it does not directly yield local policies, which are essential for decentralized execution. To address this, we propose solving the following weighted BC for each local agent $i \in \mathcal{N}$:

$$\max_{\pi_i} \mathbb{E}_{(\mathbf{s}, \mathbf{a}) \sim \mathcal{D}}\left[w^{tot*}(\mathbf{s}, \mathbf{a}) \log \pi_i(a_i|s_i)\right]. \tag{9}$$

This local WBC approach has several attractive properties. First, $w^{tot*}(\mathbf{s}, \mathbf{a})$ appears explicitly in the local policy optimization and is computed from global observations and actions. This enables local policies to be optimized with global information, ensuring consistency with the credit assignment in the multi-agent system. Furthermore, as shown in Proposition 4.3 below, the optimization of local policies through local WBC is highly consistent with the global weighted BC in 8.

**Proposition 4.3.** *Let $\pi_i^*$ be the optimal solution to the local weighted BC 9. Then $\boldsymbol{\pi}_{tot}^*(\boldsymbol{a}|\boldsymbol{s}) = \prod_{i \in \mathcal{N}} \pi_i^*(a_i|s_i)$ is also optimal for the global weighted BC in 8.*

Here we note that consistency between global and local policies is a critical aspect of centralized training with CTDE. Previous MARL approaches typically achieve this by factoring Q or V functions into local functions and training local policies based on these local functions (Rashid et al., 2020; Wang et al., 2020; Bui et al., 2024). However, in our case, there are key differences that prevent us

from employing such local values to derive local policies. Specifically, we factorize the Lagrange multipliers $\nu^{tot}$ to train the stationary distribution ratio $w^{tot}$. Although local $w$ values can be extracted from local $\nu_i$, these local $w$ values do not represent a local stationary distribution ratio and therefore cannot be used to recover local policies.

## 5 PRACTICAL ALGORITHM

Let $\mathcal{D}$ represent the offline dataset, consisting of sequences of local observations and actions gathered from a global behavior policy $\boldsymbol{\pi}_{tot}$. To train the value function $\boldsymbol{\nu}$, we construct a value network $\nu_i(s_i; \psi_\nu)$ for each local agent $i$, along with a network for each local Q-function $q_i(s_i, a_i; \psi_q)$, where $\psi_\nu$ and $\psi_q$ are learnable parameters for the local value and Q-functions. We note that the introduction and learning of the Q-functions are intended to facilitate the decomposition of the advantage function, $A_\nu^{tot}$. In our multi-agent setting, the absence of local rewards makes it difficult to directly compute local advantage functions. To overcome this challenge, we learn local Q-functions, which are then used to derive the local advantage functions. Additionally, as explained below, a MSE is optimized to ensure that the global Q-function and state-value function align properly with the global rewards.

Now, each local advantage function is then calculated as follows: The global value function and advantage function are subsequently aggregated using two mixing networks with a shared set of learnable parameters $\theta$:

$$\nu^{tot}(\mathbf{s}) = \mathcal{M}_\theta^{\mathbf{s}}[\boldsymbol{\nu}(\mathbf{s}; \psi_\nu)], \quad A_\nu^{tot}(\mathbf{s}, \mathbf{a}) = \mathcal{M}_\theta^{\mathbf{s}}[\mathbf{q}(\mathbf{s}, \mathbf{a}; \psi_q) - \boldsymbol{\nu}(\mathbf{s}; \psi_\nu)],$$

where $\mathcal{M}_\theta^{\mathbf{s}}[\cdot]$ represents a linear combination of its inputs with non-negative weights, such that $\mathcal{M}_\theta^{\mathbf{s}}[\boldsymbol{\nu}(\mathbf{s}; \psi_\nu)] = \boldsymbol{\nu}(\mathbf{s}; \psi_\nu)^\top W_\theta^{\mathbf{s}} + b_\theta^{\mathbf{s}}$, where $W_\theta^{\mathbf{s}}$ and $b_\theta^{\mathbf{s}}$ are weights of the mixing network.[1] It is important to note that $W_\theta^{\mathbf{s}}$ and $b_\theta^{\mathbf{s}}$ are generated by hyper-networks that take the global state $\mathbf{s}$ and the learnable parameters $\theta$ as inputs. In this context, we employ the same mixing network $\mathcal{M}_\theta^{\mathbf{s}}$ to combine the local values and advantages. However, our framework is flexible enough to allow the use of two different mixing networks for $\nu^{tot}$ and $A_\nu^{tot}$.

In our setting, the relationship between the global Q-function, value, and advantage functions is described in Eq. 5. Specifically, we have: $A_\nu^{tot}(\mathbf{s}, \mathbf{a}) = r(\mathcal{Z}(\mathbf{s}), \mathbf{a}) + \gamma \mathbb{E}_{\mathbf{s}' \sim P(\cdot|\mathbf{s}, \mathbf{a})}[\nu^{tot}(\mathbf{s}')] - \nu^{tot}(\mathbf{s})$. To capture this relationship, we train the Q-function by optimizing the following MSE loss:

$$\min_{\mathbf{q}} \sum_{(\mathbf{s}, \mathbf{a}, \mathbf{s}') \sim \mathcal{D}} \left( A_\nu^{tot}(\mathbf{s}, \mathbf{a}) - r(\mathcal{Z}(\mathbf{s}), \mathbf{a}) + \gamma \nu^{tot}(\mathbf{s}') - \nu^{tot}(\mathbf{s}) \right)^2.$$

This is equivalent to:

$$\min_{\psi_q} \mathcal{L}_q(\psi_q) = \sum_{(\mathbf{s}, \mathbf{a}, \mathbf{s}') \sim \mathcal{D}} \Big( \mathcal{M}_\theta^{\mathbf{s}}[\mathbf{q}(\mathbf{s}, \mathbf{a}; \psi_q) - \boldsymbol{\nu}(\mathbf{s}; \psi_\nu)]$$
$$- r(\mathcal{Z}(\mathbf{s}), \mathbf{a}) + \gamma \mathcal{M}_\theta^{\mathbf{s}'}[\boldsymbol{\nu}(\mathbf{s}'; \psi_\nu)] - \mathcal{M}_\theta^{\mathbf{s}}[\boldsymbol{\nu}(\mathbf{s}; \psi_\nu)] \Big)^2. \quad (10)$$

For the primary loss function used to train the value function, we leverage transitions from the offline dataset to approximate the objective $\widetilde{\mathcal{L}}$, resulting in the following loss function for offline training:

$$\widetilde{\mathcal{L}}(\psi_\nu, \theta) = (1-\gamma) \mathbb{E}_{\mathbf{s}_0 \sim \mathcal{D}}[\mathcal{M}_\theta^{\mathbf{s}_0}[\boldsymbol{\nu}(\mathbf{s}_0; \psi_\nu)]] + \mathbb{E}_{(\mathbf{s}, \mathbf{a}) \sim \mathcal{D}} \left[ \alpha f^* \left( \frac{\mathcal{M}_\theta^{\mathbf{s}}[\mathbf{q}(\mathbf{s}, \mathbf{a}; \psi_q) - \boldsymbol{\nu}(\mathbf{s}; \psi_\nu)]}{\alpha} \right) \right]. \quad (11)$$

As mentioned, after obtaining $(\boldsymbol{\nu}^*, \theta^*)$ by solving $\min_{\psi_\nu, \theta} \widetilde{\mathcal{L}}(\psi_\nu, \theta)$, we compute the occupancy ratio: $w_\nu^{tot*}(\mathbf{s}, \mathbf{a}) = \max \left\{ 0, f'^{-1} \left( \frac{\mathcal{M}_{\theta^*}^{\mathbf{s}}[\boldsymbol{\nu}^*(\mathbf{s})] - \mathcal{M}_{\theta^*}^{\mathbf{s}}[\mathbf{q}(\mathbf{s}, \mathbf{a}; \psi_q)]}{\alpha} \right) \right\}$. To train the local policy $\pi_i(a_i|s_i)$, we represent it using a policy network $\pi_i(a_i|s_i; \eta_i)$, where $\eta_i$ are the learnable parameters. The training process involves optimizing the following weighted behavioral cloning (BC) objective:

$$\max_{\eta_i} \quad \mathcal{L}_\pi(\eta_i) = \sum_{(\mathbf{s}, \mathbf{a}) \sim \mathcal{D}} w_\nu^{tot*}(\mathbf{s}, \mathbf{a}) \log(\pi_i(a_i|s_i; \eta_i)). \quad (12)$$

Our ComaDICE algorithm consists of two primary steps. The first step involves estimating the occupancy ratio $w^{tot*}$ from the offline dataset. The second step focuses on training the local policy

---

[1]In our experiments, we use a single-layer mixing network due to its superior performance compared to a two-layer structure, though our approach is general and can handle any multi-layer feed-forward mixing network.

by solving the weighted BC problem using $w^{tot*}$. In the first step, we simultaneously update the Q-functions $\psi_q$, the mixing network parameters $\theta$, and the value function $\psi_\nu$, aiming to minimize the mean squared error (MSE) in Eq. 10 while optimizing the main loss function in Eq. 11.

It is important to note that, in practical POMDP scenarios, the global state $\mathbf{s}$ is not directly accessible during training and is instead represented by the joint observations $\mathbf{o}$ from the agents. For notational convenience, we use the global state $\mathbf{s}$ in our formulation; however, in practice, it corresponds to the joint observation $\mathcal{Z}(\mathbf{s})$. Specifically, terms like $\rho^{\boldsymbol{\mu}_{tot}}(\mathbf{s}, \mathbf{a})$ and $\nu^{tot}(\mathbf{s})$ actually refer to $\rho^{\boldsymbol{\mu}_{tot}}(\mathbf{o}, \mathbf{a})$ and $\nu^{tot}(\mathbf{o})$, where $\mathbf{o} = \mathcal{Z}(\mathbf{s})$.

# 6 EXPERIMENTS

## 6.1 ENVIRONMENTS

We utilize three standard MARL environments: SMACv1 (Samvelyan et al., 2019), SMACv2 (Ellis et al., 2022), and Multi-Agent MuJoCo (MaMujoco) (de Witt et al., 2020), each offering unique challenges and configurations for evaluating cooperative MARL algorithms.

**SMACv1.** SMACv1 is based on Blizzard's StarCraft II. It uses the StarCraft II API and DeepMind's PySC2 to enable agent interactions with the game. SMACv1 focuses on decentralized micromanagement scenarios where each unit is controlled by an RL agent. Tasks like *2c_vs_64zg* and *5m_vs_6m* are labeled hard, while *6h_vs_8z* and *corridor* are super hard. The offline dataset, provided by Meng et al. (2023), was generated using MAPPO-trained agents (Yu et al., 2022).

**SMACv2.** In comparison to SMACv1, SMACv2 introduces increased randomness and diversity by randomizing start positions, unit types, and modifying sight and attack ranges. This version includes tasks such as *protoss*, *terran*, and *zerg*, with instances ranging from *5_vs_5* to *20_vs_23*, increasing in difficulty. Our offline dataset for SMACv2 was generated by running MAPPO for 10 million training steps and collecting 1,000 trajectories, ensuring medium quality but comprehensive coverage of the learning process. To the best of our knowledge, we are the first to explore SMACv2 in offline MARL, whereas most prior work has used this environment in online settings.

**MaMujoco.** MaMujoco serves as a benchmark for continuous cooperative multi-agent robotic control. Derived from the single-agent MuJoCo control suite in OpenAI Gym (Brockman et al., 2016), it presents scenarios where multiple agents within a single robot must collaborate to achieve tasks. The tasks include *Hopper-v2*, *Ant-v2*, and *HalfCheetah-v2*, with instances labeled as *expert*, *medium*, *medium-replay*, and *medium-expert*. The offline dataset was created by (Wang et al., 2022b) using the HAPPO method (Wang et al., 2022a).

## 6.2 BASELINES

We consider the following baselines, which represent either standard or state-of-the-art (SOTA) methods for offline MARL: (i) **BC** (Behavioral Cloning); (ii) **BCQ** (Batch-Constrained Q-learning) (Fujimoto et al., 2019) – an offline RL algorithm that constrains the policy to actions similar to those in the dataset to reduce distributional shift, adapted for offline MARL settings; (iii) **CQL** (Conservative Q-Learning) (Kumar et al., 2020) – a method that stabilizes offline Q-learning by penalizing out-of-distribution actions, ensuring conservative value estimates; (iv) **ICQ** (Implicit Constraint Q-learning) (Yang et al., 2021) – an approach using importance sampling to manage out-of-distribution actions in multi-agent settings; (v) **OMAR** (Offline MARL with Actor Rectification) (Pan et al., 2022) – a method combining CQL with optimization techniques to ensure the global validity of local regularizations, promoting cooperative behavior; (vi) **OMIGA** (Offline MARL with Implicit Global-to-Local Value Regularization) (Wang et al., 2022b) – a SOTA method that transforms global regularizations into implicit local ones, optimizing local policies with global insights; (vii) **OptDICE** - a naive extension of the OptDICE algorithm Lee et al. (2021) to multi-agent settings where the global value function are directly learned without value factorization; and (viii) **AlberDICE** Matsunaga et al. (2023) - an offline MARL algorithm which also leverages the DICE framework to address the OOD.

We used experimental results contributed by the authors of OMIGA (Wang et al., 2022b) as our baselines. They provided both the results and source code for all the baseline methods. This source

| Instances | | BC | BCQ | CQL | ICQ | OMAR | OMIGA | OptDICE | AlberDICE | ComaDICE (ours) |
|---|---|---|---|---|---|---|---|---|---|---|
| 2c_vs_64zg | poor | 0.0 ± 0.0 | 0.0 ± 0.0 | 0.0 ± 0.0 | 0.0 ± 0.0 | 0.0 ± 0.0 | 0.0 ± 0.0 | 0.0 ± 0.0 | 0.0 ± 0.0 | **0.6 ± 1.3** |
| | medium | 1.9 ± 1.5 | 2.5 ± 3.6 | 2.5 ± 3.6 | 1.9 ± 1.5 | 1.2 ± 1.5 | 6.2 ± 5.6 | 1.0 ± 1.5 | 1.6 ± 1.6 | **8.8 ± 7.0** |
| | good | 31.2 ± 9.9 | 35.6 ± 8.8 | 44.4 ± 13.0 | 28.7 ± 4.6 | 28.7 ± 9.1 | 40.6 ± 9.5 | 37.5 ± 3.1 | 42.2 ± 6.4 | **55.0 ± 1.5** |
| 5m_vs_6m | poor | 2.5 ± 1.3 | 1.2 ± 1.5 | 1.2 ± 1.5 | 1.2 ± 1.5 | 0.6 ± 1.2 | **6.9 ± 1.2** | 0.0 ± 0.00 | 0.0 ± 0.0 | 4.4 ± 4.2 |
| | medium | 1.9 ± 1.5 | 1.2 ± 1.5 | 2.5 ± 1.2 | 1.2 ± 1.5 | 0.6 ± 1.2 | 2.5 ± 3.1 | 0.0 ± 0.00 | 3.1 ± 0.0 | **7.5 ± 2.5** |
| | good | 2.5 ± 2.3 | 1.9 ± 2.5 | 1.9 ± 1.5 | 3.8 ± 2.3 | 3.8 ± 1.2 | 6.9 ± 1.2 | 7.3 ± 3.9 | 3.9 ± 1.4 | **8.1 ± 3.2** |
| 6h_vs_8z | poor | 0.0 ± 0.0 | 0.0 ± 0.0 | 0.0 ± 0.0 | 0.0 ± 0.0 | 0.0 ± 0.0 | 0.0 ± 0.0 | 0.0 ± 0.0 | 1.0 ± 1.5 | **1.9 ± 3.8** |
| | medium | 1.9 ± 1.5 | 1.9 ± 1.5 | 1.9 ± 1.5 | 1.9 ± 1.5 | 2.5 ± 1.2 | 1.2 ± 1.5 | 0.0 ± 0.0 | 2.3 ± 2.6 | **3.1 ± 2.0** |
| | good | 8.8 ± 1.2 | 8.8 ± 3.6 | 7.5 ± 1.5 | 9.4 ± 2.0 | 0.6 ± 1.3 | 5.6 ± 3.6 | 0.0 ± 0.0 | 0.0 ± 0.0 | **11.2 ± 5.4** |
| corridor | poor | 0.0 ± 0.0 | 0.0 ± 0.0 | 0.0 ± 0.0 | **0.6 ± 1.3** | 0.0 ± 0.0 | 0.0 ± 0.0 | 0.0 ± 0.0 | 0.0 ± 0.0 | **0.6 ± 1.3** |
| | medium | 15.0 ± 2.3 | 23.1 ± 1.5 | 14.4 ± 1.5 | 22.5 ± 3.1 | 11.9 ± 2.3 | 23.8 ± 5.1 | 19.8 ± 2.9 | 9.4 ± 6.8 | **27.3 ± 3.4** |
| | good | 30.6 ± 4.1 | 42.5 ± 6.4 | 5.6 ± 1.2 | 42.5 ± 6.4 | 3.1 ± 0.0 | 41.9 ± 6.4 | 39.6 ± 5.3 | 43.1 ± 6.4 | **48.8 ± 2.5** |

Table 1: Comparison of average winrates for ComaDICE and baselines on SMACv1 tasks.

| Instances | | BC | BCQ | CQL | ICQ | OMAR | OMIGA | OptDICE | AlberDICE | ComaDICE (ours) |
|---|---|---|---|---|---|---|---|---|---|---|
| Protoss | 5_vs_5 | 36.9±8.7 | 16.2±2.3 | 10.0±4.1 | 36.9±9.1 | 21.2±4.1 | 33.1±5.4 | 10.8±1.2 | 12.6±0.9 | **46.2±6.1** |
| | 10_vs_10 | 36.2±10.6 | 9.4±5.6 | 26.2±7.6 | 28.1±6.6 | 13.8±7.0 | 40.0±10.7 | 9.5±0.8 | 11.8±0.9 | **50.6±8.7** |
| | 10_vs_11 | 19.4±4.6 | 10.0±4.1 | 10.6±5.4 | 12.5±4.4 | 12.5±3.4 | 16.2±6.1 | 10.0±0.5 | 9.8±0.3 | **20.0±4.2** |
| | 20_vs_20 | 37.5±4.4 | 6.2±2.0 | 11.9±4.1 | 32.5±8.1 | 23.8±2.5 | 36.2±5.1 | 10.0±2.0 | 10.1±0.6 | **47.5±7.8** |
| | 20_vs_23 | **13.8±1.5** | 1.2±1.5 | 0.0±0.0 | 12.5±5.6 | 11.2±7.8 | 12.5±8.1 | 8.1±1.4 | 8.8±0.8 | 13.8±5.8 |
| Terran | 5_vs_5 | 30.0±4.2 | 12.5±6.2 | 9.4±7.9 | 23.1±5.8 | 14.4±4.7 | 28.1±4.4 | 6.4±1.1 | 8.1±1.4 | **30.6±8.2** |
| | 10_vs_10 | 29.4±5.8 | 6.9±6.1 | 9.4±5.6 | 16.9±5.8 | 15.0±4.6 | 29.4±3.2 | 6.0±1.6 | 8.2±1.0 | **32.5±5.8** |
| | 10_vs_11 | 16.2±3.6 | 3.8±4.6 | 7.5±6.4 | 5.0±4.2 | 9.4±5.6 | 12.5±5.2 | 4.8±1.2 | 6.2±0.9 | **19.4±5.4** |
| | 20_vs_20 | 26.2±10.4 | 5.0±3.2 | 10.6±4.2 | 15.6±3.4 | 7.5±7.3 | 21.9±4.4 | 6.3±1.8 | 5.9±1.2 | **29.4±3.8** |
| | 20_vs_23 | 4.4±4.2 | 0.0±0.0 | 0.0±0.0 | 7.5±6.1 | 5.0±4.2 | 4.4±2.5 | 4.4±0.7 | 3.9±0.8 | **9.4±5.2** |
| Zerg | 5_vs_5 | 26.9±10.0 | 14.4±4.2 | 14.4±5.8 | 18.8±7.1 | 13.8±6.1 | 21.9±5.9 | 8.2±1.8 | 9.5±0.8 | **31.2±7.7** |
| | 10_vs_10 | 25.0±2.8 | 5.6±4.6 | 5.6±4.6 | 15.6±7.4 | 19.4±2.3 | 23.8±6.4 | 7.8±1.0 | 8.5±0.3 | **33.8±11.8** |
| | 10_vs_11 | 13.8±4.7 | 9.4±5.2 | 6.2±4.4 | 10.6±6.7 | 10.6±3.8 | 13.8±6.7 | 7.2±0.7 | 9.1±0.5 | **19.4±3.6** |
| | 20_vs_20 | 8.1±1.5 | 2.5±1.2 | 1.2±1.5 | 10.0±7.8 | **12.5±4.4** | 10.0±2.3 | 7.3±0.7 | 8.3±0.5 | 9.4±6.2 |
| | 20_vs_23 | 7.5±3.2 | 0.6±1.3 | 1.2±1.5 | 7.5±3.2 | 3.8±2.3 | 4.4±4.2 | 7.1±1.2 | 8.8±0.5 | **11.2±4.2** |

Table 2: Comparison of win rates for ComaDICE and baselines across SMACv2 tasks.

code was also employed to run these baselines for the SMACv2 environment. All hyperparameters were kept at their default settings, and each experiment was conducted with *five different random seeds* to ensure robustness and reproducibility of the results.

## 6.3 MAIN COMPARISON

We now present a comprehensive evaluation of our proposed algorithm, ComaDICE, against several baseline methods in offline MARL. The baselines selected for comparison include both standard and SOTA approaches, providing a robust benchmark to assess the effectiveness of ComaDICE.

Our evaluation focuses on two primary metrics: returns and winrates. Returns are the average rewards accumulated by the agents across multiple trials, providing a measure of policy effectiveness. Winrates, applicable in competitive environments such as SMACv1 and SMACv2, indicate the success rate of agents against opponents, reflecting the algorithm's robustness in adversarial settings.

The experimental results, summarized in Tables 1-3, demonstrate that ComaDICE consistently achieves superior performance compared to baseline methods across a range of scenarios. Notably, ComaDICE excels in complex tasks, highlighting its ability to effectively manage distributional shifts in challenging environments.

## 6.4 ABLATION STUDY - IMPACT OF THE REGULARIZATION PARAMETER ALPHA

We investigate how varying the regularization parameter alpha ($\alpha$) affects the performance of our ComaDICE algorithm. The parameter $\alpha$ is crucial for balancing the trade-off between maximizing rewards and penalizing deviations from the offline dataset's distribution. We conducted experiments with $\alpha$ values ranging from $\{0.01, 0.1, 1, 10, 100\}$, evaluating performance using average winrates across all the SMACv2 tasks and average returns across all the MaMujoco tasks. These results, illustrated in Figure 1, highlight the sensitivity of ComaDICE to different $\alpha$ values. In particular, we

| Instances | | BCQ | CQL | ICQ | OMIGA | OptDICE | AlberDICE | ComaDICE (ours) |
|---|---|---|---|---|---|---|---|---|
| Hopper | expert | 77.9 ± 58.0 | 159.1 ± 313.8 | 754.7 ± 806.3 | 859.6 ± 709.5 | 655.9 ± 120.1 | 844.6 ± 556.5 | **2827.7 ± 62.9** |
| | medium | 44.6 ± 20.6 | 401.3 ± 199.9 | 501.8 ± 14.0 | **1189.3 ± 544.3** | 204.1 ± 41.9 | 216.9 ± 35.3 | 822.6 ± 66.2 |
| | m-replay | 26.5 ± 24.0 | 31.4 ± 15.2 | 195.4 ± 103.6 | 774.2 ± 494.3 | 257.8 ± 55.3 | 419.2 ± 243.5 | **906.3 ± 242.1** |
| | m-expert | 54.3 ± 23.7 | 64.8 ± 123.3 | 355.4 ± 373.9 | 709.0 ± 595.7 | 400.9 ± 132.5 | 515.1 ± 303.4 | **1362.4 ± 522.9** |
| Ant | expert | 1317.7 ± 286.3 | 1042.4 ± 2021.6 | 2050.0 ± 11.9 | 2055.5 ± 1.6 | 1717.2 ± 27.0 | 1896.8 ± 33.7 | **2056.9 ± 5.9** |
| | medium | 1059.6 ± 91.2 | 533.9 ± 1766.4 | 1412.4 ± 10.9 | 1418.4 ± 5.4 | 1199.0 ± 26.8 | 1304.3 ± 2.6 | **1425.0 ± 2.9** |
| | m-replay | 950.8 ± 48.8 | 234.6 ± 1618.3 | 1016.7 ± 53.5 | 1105.1 ± 88.9 | 869.4 ± 62.6 | 1042.8 ± 80.8 | **1122.9 ± 61.0** |
| | m-expert | 1020.9 ± 242.7 | 800.2 ± 1621.5 | 1590.2 ± 85.6 | 1720.3 ± 110.6 | 1293.2 ± 183.1 | 1780.0 ± 23.6 | **1813.9 ± 68.4** |
| Half Cheetah | expert | 2992.7 ± 629.7 | 1189.5 ± 1034.5 | 2955.9 ± 459.2 | 3383.6 ± 552.7 | 2601.6 ± 461.9 | 3356.4 ± 546.9 | **4082.9 ± 45.7** |
| | medium | 2590.5 ± 1110.4 | 1011.3 ± 1016.9 | 2549.3 ± 96.3 | **3608.1 ± 237.4** | 305.3 ± 946.8 | 522.4 ± 315.5 | 2664.7 ± 54.2 |
| | m-replay | -333.6 ± 152.1 | 1998.7 ± 693.9 | 1922.4 ± 612.9 | 2504.7 ± 83.5 | -912.9 ± 1363.9 | 440.0 ± 528.0 | **2855.0 ± 242.2** |
| | m-expert | 3543.7 ± 780.9 | 1194.2 ± 1081.0 | 2834.0 ± 420.3 | 2948.5 ± 518.9 | -2485.8 ± 2338.4 | 2288.2 ± 759.5 | **3889.7 ± 81.6** |

Table 3: Average returns for ComaDICE and baselines on MaMuJoCo benchmarks.

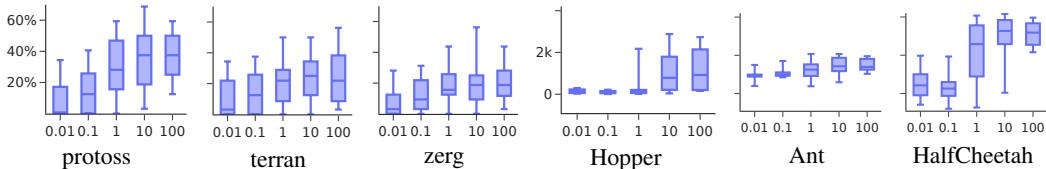

Figure 1: Impact of regularization parameter $\alpha$ on performance in different environments.

observe that ComaDICE achieves optimal performance when $\alpha$ is around 10, suggesting that the stationary distribution regularizer plays a essential role in the success of our algorithm.

In our appendix, we provide additional ablation studies to analyze the performance of our algorithm using different forms of f-divergence functions, as well as comparisons between 1-layer and 2-layer mixing network structures. The appendix also includes proofs of the theoretical claims made in the main paper, details of our experimental settings, and other experimental information.

# 7 CONCLUSION, FUTURE WORK AND BROADER IMPACTS

**Conclusion.** In this paper, we propose ComaDICE, a principled framework for offline MARL. Our algorithm incorporates a stationary distribution shift regularizer into the standard MARL objective to address the conventional distribution shift issue in offline RL. To facilitate training within a CTDE framework, we decompose both the global value and advantage functions using a mixing network. We demonstrate that, under our mixing architecture, the main objective function is concave in the value function, which is crucial for ensuring stable and efficient training. The results of this training are then utilized to derive local policies through a weighted BC approach, ensuring consistency between global and local policy optimization. Extensive experiments on SOTA benchmark tasks, including SMACv2, show that ComaDICE outperforms other baseline methods.

**Limitations and Future Work:** There are some limitations that are not addressed within the scope of this paper. For instance, we focus solely on cooperative learning, leaving open the question of how the approach would perform in cooperative-competitive settings. Additionally, in our training objective, the DICE term is designed to reduce the divergence between the learning policy and the behavior policy. As a result, the performance of the algorithm is heavily dependent on the quality of the behavior policy. Furthermore, our algorithm, like other baselines, still requires a large amount of data to achieve desirable learning outcomes. Improving sample efficiency would be another valuable area for future research.

**Broader Impacts:** Developing an offline MARL algorithm with a stationary distribution shift regularizer can enhance performance in costly real-time tasks like robotics, autonomous driving, and healthcare. It also enables safer exploration and broader adoption in high-stakes settings. However, reliance on the behavior policy means flawed or biased data could degrade performance, reinforcing biases or suboptimal behaviors. Additionally, the algorithm, like any AI systems, risks unintended misuse in surveillance or military applications, where multi-agent systems could manipulate environments without proper oversight.

## ACKNOWLEDGMENT

This work is supported by the Lee Kong Chian Fellowship awarded to Tien Mai.

## ETHICAL STATEMENT

Our work introduces ComaDICE, a framework for offline MARL, aimed at improving training stability and policy optimization in complex multi-agent environments. While this research has significant potential for positive applications, particularly in domains such as autonomous systems, resource management, and multi-agent simulations, it is crucial to address the ethical implications and risks associated with this technology.

The deployment of reinforcement learning systems in real-world, multi-agent settings raises concerns about unintended behaviors, especially in safety-critical domains. If the policies learned by ComaDICE are applied without proper testing and validation, they may lead to undesirable or harmful outcomes, especially in areas such as autonomous driving, healthcare, or robotics. Additionally, bias in the training data or simulation environments could result in suboptimal policies that unfairly impact certain agents or populations, potentially leading to ethical concerns regarding fairness and transparency.

To mitigate these risks, we emphasize the need for extensive testing and validation of policies generated using ComaDICE, particularly in real-world environments where the consequences of errors could be severe. It is also essential to ensure that the datasets and simulations used in training are representative, unbiased, and carefully curated. We encourage practitioners to use human oversight and collaborate with domain experts to ensure that ComaDICE is applied responsibly, particularly in high-stakes settings.

## REPRODUCIBILITY STATEMENT

In order to facilitate reproducibility, we have submitted the source code for ComaDICE, along with the datasets utilized to produce the experimental results presented in this paper (all these will be made publicly available if the paper gets accepted). Additionally, in the appendix, we provide details of our algorithm, including key implementation steps and details needed to replicate the results. The hyper-parameter settings for all experiments are also included to ensure that others can reproduce the findings under the same experimental conditions. We invite the research community to explore and apply the ComaDICE framework in various environments to further validate and expand upon the results reported in this work.

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
