# OpenReview forum: "ComaDICE: Offline Cooperative Multi-Agent Reinforcement Learning with Stationary Distribution Shift Regularization"
_ICLR.cc/2025/Conference — ICLR 2025 Poster_

### Official Review · Reviewer_XXjk · 2024-11-01

**Soundness:** 3
**Presentation:** 3
**Contribution:** 3
**Rating:** 8
**Confidence:** 4

**Summary:**

This work introduces ComaDICE (offline Cooperative MARL with DICE), an approach for offline cooperative multi-agent reinforcement learning (RL) that leverages the DICE method. ComaDICE formulates the offline cooperative multi-agent RL problem as a constrained optimization and employs a DICE-based method to compute a global Lagrangian multiplier, $\nu^{tot}$. Given the large state and action spaces typical in multi-agent RL, practical optimization of $\nu^{tot}$ may not be feasible. To address this challenge, ComaDICE employs value function decomposition to decompose the global Lagrangian multiplier into individual Lagrangian multipliers for each agent.

**Strengths:**

This work provides a comprehensive performance evaluation through experiments conducted on an extensive set of benchmarks, including the challenging multi-agent RL benchmark SMAC-v2. The experimental results indicate that the proposed approach shows superior performance relative to the baselines considered in the manuscript.

**Weaknesses:**

The primary concern regarding this work is its novelty compared to previous work.

Specifically, AlberDICE by Matsunaga et al. (2023) [A] conveys a similar idea, i.e. adoption of DICE for offline cooperative multi-agent RL. The key difference of ComaDICE from AlberDICE appears to be employing individual Langrangian multipliers $\nu_i$ and the mixing network for value function decomposition: while AlberDICE utilizes a simple yet principled resampling method for obtaining $\nu_i$, ComaDICE employs the value factorization as discussed in Section 4.2, which requires the reliance on the additional mixing network $\mathcal{M}_\theta$ further necessitates additional training.

Although AlberDICE is briefly mentioned in Section 2, the paper does not adequately discuss the theoretical or empirical advantages of adopting the value decomposition instead of the resampling approach. In addition, the AlberDICE paper presents a simple multi-agent task where value function decomposition leads to a substantial performance loss.

The absence of such a comparison significantly weakens the perceived contribution of this study, as it fails to establish a clear improvement of differentiation from previous work. In this sense,
(1) A detailed discussion comparing ComaDICE and AlberDICE should be added,
(2) AlberDICE should be included as a baseline in all experiments,
(3) ComaDICE should be tested on the XOR game in the AlberDICE paper


[A] Matsunaga et al., “AlberDICE: Addressing Out-Of-Distribution Joint Actions in Offline Multi-Agent RL via Alternating Stationary Distribution Correction Estimation.”, NeurIPS 2023.

[Minor comments]
In Sections 6.1 and 6.3, SMACv1 environment is discussed; however, corresponding experimental results for SMACv1 are not included in the main manuscript. It would be more appropriate to relocate these discussions to the appendix and direct readers there for further details, or incorporate the result on SMACv1 to the main manuscript, potentially replacing some of the redundant results on SMACv2 in Table 1 or Figure 1.

**Questions:**

Q1. Do the authors assume that $s$ can be sufficiently represented from $o$?
Notations for states and observation are confusingly used. For example, in line 234 and 237, $\nu(s)$ and $q(s,a)$ are defined as a collection of functions that requires individual observations,$\nu_i(o_i)$ and $q(o_i,a_i)$ respectively If so, the assumption should be explicitly clarified in the POMDP description.

Q2. Figure 2 lacks information on which task was selected for each benchmark. (e.g. 5_vs_5 or 10_vs_10 in protoss, or expert or medium data quality in Hopper) Could you clarify?

---

> ### Author Response · Authors · 2024-11-22
> **We thank the reviewers for the feedback!**
>
> We thank the reviewer for the comments. Please find below our responses to your concerns.
>
> >Although AlberDICE is briefly mentioned in Section 2, the paper does not adequately discuss the theoretical or empirical advantages of adopting the value decomposition instead of the resampling approach. In addition, the AlberDICE paper presents a simple multi-agent task where value function decomposition leads to a substantial performance loss.
>
> We thank the reviewer for the insightful comments, which we greatly value and have made every effort to address. Here we would like to clarify that the key distinction between our algorithm and AlberDICE lies in the learning approach: our method learns the occupancy ratio at the global level, utilizing centralized learning and decentralized execution. In contrast, AlberDICE focuses on learning individual occupancy ratios, which restricts its ability to capture the interconnections between local agents. We have added clarifications on Page 3 to emphasize these differences.
>
> >(1) A detailed discussion comparing ComaDICE and AlberDICE should be added, (2) AlberDICE should be included as a baseline in all experiments, (3) ComaDICE should be tested on the XOR game in the AlberDICE paper
>
> In the updated version, we have included comparisons with AlberDICE and OptDICE and updated our experiments accordingly. Additionally, we tested ComaDICE on the XOR game, with the results reported in Section B8 of the appendix. These results demonstrate that, for this simple game, ComaDICE achieves a similar best policy value as AlberDICE.
>
> >In Sections 6.1 and 6.3, SMACv1 environment is discussed; however, corresponding experimental results for SMACv1 are not included in the main manuscript...
>
> Thank you for the suggestion. We have now moved the results for SMACV1 to the main paper.
>
> > Do the authors assume that $s$ can be sufficiently represented from $o$? Notations for states and observation are confusingly used.
>
> In the POMDP setting, global states are not fully accessible and are instead represented by the joint observations from local agents. For simplicity, we use global state notation, but it actually refers to the corresponding joint observations. We have added a clarification on Page 7.
>
>
> > Figure 2 lacks information on which task was selected for each benchmark. (e.g. 5_vs_5 or 10_vs_10 in protoss, or expert or medium data quality in Hopper) Could you clarify?
>
>  For this figure, we used all the tasks and calculated the average win rates or returns for comparison. We have added a sentence to clarify this point.
>
> **We hope our revisions and our response  address the reviewers’ concerns and further clarify our contributions. If there are any additional questions or comments, we would be happy to address them.**

---

> > ### Author Response · Authors · 2024-11-26
> >
> > Dear Reviewer XXjk,
> >
> > As the rebuttal period is coming to an end, we kindly ask if you could take a moment to review our responses to see if they address your concerns.
> >
> > Once again, we greatly appreciate your comments, which have been invaluable in helping us improve the paper. We hope to engage in further discussions and hear more feedback from you.
> >
> > All the best,
> > The Authors

---

### Official Review · Reviewer_3fo7 · 2024-11-02

**Soundness:** 3
**Presentation:** 2
**Contribution:** 3
**Rating:** 6
**Confidence:** 4

**Summary:**

In this paper, ComaDICE algorithm is proposed for offline multi-agent reinforcement learning, extending the DICE method to multi-agent scenarios and using value function decomposition and policy extraction methods, the idea is novel and the experimental results show the potential of the method.

**Strengths:**

* The introduction of the DICE algorithm within the domain of multi-agent reinforcement learning represents a significant contribution characterized by its novelty and innovative approach.
* The experimental settings are comprehensive, demonstrating the potential of the proposed algorithm effectively.

**Weaknesses:**

* The contribution emphasizes the value decomposition and claims to prove the equivalence of the local policy product to the globally optimal policy, but the paper is too repetitive and superficial in its exposition of the decomposition method, neither explicitly defining the local subtasks and their corresponding policies, nor the relationship between them, nor providing a complete proof of this equivalence.
* The experimental results for hybrid networks contradict mainstream research findings (single-layer outperforms double-layers), a phenomenon that hints at a possible fundamental flaw in the application of the DICE methodology to MARL, but the paper lacks an in-depth discussion of this.

**Questions:**

* The discussion of the DICE method and its goals in the Related Work and Preliminary Knowledge section is not sufficiently in-depth to clearly locate the innovations of this paper with respect to existing work.
* The proof section of DICE skips some key steps,  has a confusingly organized derivation and notation that does not clearly state the purpose and rationale for each step of mathematical transformation. Interpretation between Lagrange functions and offline reinforcement learning problem formulation is in lack.
* The paper fails to explore the common sparse reward scenario in offline reinforcement learning, and the analysis of the quality requirements of behavioral strategies is insufficient, limiting the feasibility of the method in practical applications.

---

> ### Author Response · Authors · 2024-11-22
> **We thank the reviewers for the feedback!**
>
> We thank the reviewer for the comments. Please find below our responses to your concerns.
>
> > The experimental results for hybrid networks contradict mainstream research findings (single-layer outperforms double-layers) ...
>
> The low performance of the 2-layer mixing network clearly indicates that, in our offline setting, the 2-layer structure is overly complex for capturing the interdependencies between agents, leading to overfitting. Increasing the amount of offline data might reveal more information and potentially improve the performance of the 2-layer mixing network. However, this is not practical, as it would require significantly more storage and make training computationally expensive and infeasible.
> We have added an additional discussion to Section B.6 in the appendix to clarify this point. If our explanation remains unclear or if you have any further questions, we would be happy to provide additional clarification.
>
> >The discussion of the DICE method and its goals in the Related Work and Preliminary Knowledge section is not sufficiently in-depth to clearly locate the innovations of this paper with respect to existing work.
>
> In the updated version, we have made every effort to position our contributions in relation to existing works and techniques. The updates have been highlighted in magenta for clarity. If the reviewer has specific points that require more in-depth discussion or analysis, we would be happy to explore them further and provide additional clarification.
>
> > The proof section of DICE skips some key steps, has a confusingly organized derivation and notation that does not clearly state the purpose and rationale for each step of mathematical transformation. ...
>
> Due to space constraints, we had to omit some steps in our derivations and instead referenced prior work on which our approach builds. Specifically, the derivations up to Proposition 4.1 closely follow those in the OptDICE paper for single-agent RL. To clarify this, we have added some lines after Proposition 4.1 to acknowledge this fact.
>
> > The paper fails to explore the common sparse reward scenario in offline reinforcement learning, and the analysis of the quality requirements of behavioral strategies is insufficient, limiting the feasibility of the method in practical applications.
>
> We acknowledge that our work did not address the sparse reward scenario or the quality of the behavioral policy. These are indeed important aspects; however, tackling these issues would require significant additional effort, which we believe warrants a separate paper. We will acknowledge this as a limitation of our work and highlight it as a promising direction for future research.
>
> **We hope our revisions and our response  address the reviewers’ concerns and further clarify our contributions. If there are any additional questions or comments, we would be happy to address them**

---

> > ### Comment · Reviewer_3fo7 · 2024-11-24
> >
> > Thank you for addressing my previous concerns. I appreciate the effort put into the additional content, which has strengthened the manuscript significantly.
> >
> > However, I have some observations regarding the formulation section of ComaDICE. The current presentation lacks a professional and cohesive mathematical structure. As it stands, the section appears more like a draft or homework exercise rather than a polished and rigorous exposition expected in a professional context. I recommend reorganizing this section to present the derivations and explanations with greater clarity and structure, adhering to standard mathematical writing conventions.
> >
> > Additionally, the notations in this section require refinement for consistency and precision. For instance, the notation for the occupancy measure, $\rho^{\mathbf{\pi}_{\text{tot}}}$, appears to be inconsistently defined. In Equations (1) and (2), it is associated with the support space $\mathcal{O} \times \mathcal{A}$, whereas starting from Equation (3), it shifts to the support space $\mathcal{S} \times \mathcal{A}$. This inconsistency may lead to confusion and undermines the readability of the paper. I strongly encourage the authors to clarify and standardize the usage of notations throughout the manuscript to ensure precision and ease of understanding for the reader.
> >
> > Addressing these points will significantly enhance the clarity and professionalism of the paper, making it more accessible and impactful to the audience. Thank you for considering these suggestions.

---

> ### Author Response · Authors · 2024-11-24
> **We thank the reviewer for the additional feedback**
>
> > However, I have some observations regarding the formulation section of ComaDICE. The current presentation lacks a professional and cohesive mathematical structure. As it stands, the section appears more like a draft or homework exercise rather than a polished and rigorous exposition expected in a professional context. I recommend reorganizing this section to present the derivations and explanations with greater clarity and structure, adhering to standard mathematical writing conventions.
>
> We sincerely thank the reviewer for their valuable feedback, which we greatly appreciate. Regarding the exposition, we have made every effort to present our formulation clearly. However, due to space constraints, some details have been omitted, and we have provided references to relevant papers for further information. If the reviewer could kindly point out specific parts of the section that require revision or improvement, we would be more than happy to address them and make the necessary revisions to enhance the clarity and quality of our work.
>
> > Additionally, the notations in this section require refinement for consistency and precision. For instance, the notation for the occupancy measure ...
>
> We thank the reviewer for their valuable suggestion, which we have carefully considered. In response, **we have now updated the notation to ensure consistency throughout the paper.** Specifically, we now use global state notation consistently, with a clarification that, in practice, global state information is not fully available. Instead, we rely on joint observations to address this limitation.
>
> Once again, we sincerely appreciate the reviewer’s additional feedback, which has helped us further improve the paper. We hope our responses and updates adequately address your remaining concerns.
>
> **Should you have any further questions or require additional clarification, we would be more than happy to provide further explanations or revisions.**

---

> > ### Comment · Reviewer_3fo7 · 2024-11-26
> >
> > Thank the authors for your valuable responses. Regarding Section 4, while the derivation itself is interesting, I believe this section could benefit from a clearer structure. For example, for Section 4.1, I suggest explicitly defining the problem at the outset and clearly highlighting the differences between your work and established approaches. For the complete proof, I think it is still essential for completence of this work, which you can add in the appendix.
> > Some minor points:
> > * Why is \mu_{tot} not bold, given that it is also a vector?
> > * Some text formatting seems strange; for example, why are terms like "s.t." and \sum in Eq. (3)) is bold?
> > * Lines 201 and 215 appear to serve the similar function, but line 201 is not labeled, whereas line 215 is labeled as Eq. (6). Could you clarify or adjust for consistency? If line 201 is not important in this section, I recommend you to keep it as inline equation to highlight the problem you are dealing with.
> > * Some equations lack proper punctuation, such as commas or periods. Please ensure these are included for grammatical and stylistic accuracy. E.g., comma in Eq (3) should be a period; Eq (4) lacks a comma; Eq (6) lacks a period, etc.

---

> > > ### Author Response · Authors · 2024-11-26
> > >
> > > We thank the reviewer for the additional feedback, which is valuable in helping us further improve the paper. We have updated our manuscript to incorporate the following changes:
> > >
> > > - Cited more relevant works and added discussions to provide more context.
> > > - Revised some notations as suggested, e.g., changed $\mu^{tot}$ to bold font for consistency with $\pmb{\pi}^{tot}$, modified the equation in line 201 to align with the text, and added periods and commas for improved readability.
> > >
> > > Due to space constraints, we were unable to include many additional equations and explanations in the main paper. Instead, we have added references in relevant sections to clarify our derivations.
> > >
> > > We hope that this updated version addresses your remaining concerns. If you have any further suggestions, we would be happy to incorporate them into our paper.

---

### Official Review · Reviewer_kymz · 2024-11-02

**Soundness:** 3
**Presentation:** 2
**Contribution:** 3
**Rating:** 6
**Confidence:** 5

**Summary:**

The paper proposes ComaDICE, a stationary distribution correction estimation approach to addressing OOD states and actions in Offline MARL. The derivation starts with the LP formulation of RL in the joint state-action space, which results in a concave objective for learning state-value functions similar to OptiDICE [2]. The objective also uses value decomposition with a monotonic mixing network for the advantage function similar to OMIGA [3]. ComaDICE is evaluated on SMACv2 and MaMuJoCo and shows comparable performance with baselines such as BC and OMIGA.

**Strengths:**

It is known that every MDP has a deterministic optimal policy, which, if extended to the multi-agent setting, can be factorized into independent policies. A very optimistic reading of the paper would interpret that the decomposition procedure is actually learning optimal value functions over decentralized policies. If this interpretation is correct, the ComaDICE presents a very scalable and principled offline MARL algorithm for learning decentralized value functions and policies without any restrictive IGM assumption as in previous work such as ICQ. However, it is worth noting that this is more of a statement of the potential of the paper rather than a strength of its current version, as these points are not specifically addressed in the current draft.

Furthermore, ComaDICE can be more scalable in comparison to AlberDICE [1] which requires alternating optimization.

**Weaknesses:**

### Problem Statement
It is not clear what the main problem is that ComaDICE is solving. It is briefly mentioned in the introduction that OOD states and actions are a problem in offline MARL. However, this was addressed in detail by other work such as CFCQL/OMIGA/AlberDICE mentioned in the Related Work (especially AlberDICE [1] which is the most similar). For instance, AlberDICE considers some coordination problems (XOR/Bridge/etc.) where OOD joint actions may be common, these is no consideration of these settings as well as comparison to AlberDICE both algorithmically and empirically. Thus, it is not entirely clear from the current draft of the paper what the main problem is that ComaDICE is solving, and if it is indeed OOD actions, which part of the algorithm in particular is alleviating this.

### Novelty
The derivation is based on OptiDICE [2] but this is not explicitly mentioned, which is misleading.
For instance, Proposition 4.1 seems equivalent to Proposition 1 of OptiDICE. Furthermore, an extension of OptiDICE to solve Offline MARL was considered in AlberDICE [1] so it is not clear why ComaDICE should be preferred over AlberDICE. Furthermore, the final algorithm closely resembles OMIGA [3].

### Algorithm
It is unclear what the purpose of value decomposition in learning the Q functions is. It seems the advantage can be computed by the learned state-value function $\nu$ and run WBC (as mentioned in Appendix C of AlberDICE[1]). Also, Line 346 defines $A_\nu^{tot}$ as the sum of reward and state-value functions.

### Lack of Relevant Baselines
AlberDICE and OptiDICE are missing as the main baselines, as well as CFCQL which addresses OOD actions but in a different manner. These are all mentioned in the Related Work section but not compared.

### Writing
The paper is generally not well-written. All of the aforementioned weaknesses of the paper should be addressed in detail and the writing should not raise any of these concerns.

**Questions:**

1. What is the main purpose for the value decomposition and learning individual Q-functions?

1. Related to the first question, what is the purpose of Theorem 4.2, if the original loss is also concave? If we use the mixing functions, does $ \mathcal{\tilde L}$ still find the global optimum?

1. Can ComaDICE solve the XOR game and Bridge mentioned in AlberDICE?

1. Related to “Strengths”, can solving ComaDICE learn the global optimum in the underlying MDP even with the mixing network?


1. Is the Individual Global Max (IGM) assumption required in order to introduce the mixing network? In other words, does ComaDICE assume that the underlying optimal Q function assume IGM?

1. What is the main difference in the final algorithm with OMIGA [3] ?

1. Why are the notations mixed between using joint observations $\mathbf{o}$ and $s$? Is the setting a Dec-POMDP or a specialized setting where the joint observations constitute the state?

1. Please write in a different color during the rebuttal which part is the novel part. In particular, which part of the proofs are a novel contribution and which part is coming from OptiDICE [2]

1. Please address both my comments in the Strengths and Weaknesses section in detail.


### References
[1] AlberDICE: Addressing Out-Of-Distribution Joint Actions in Offline Multi-Agent RL via Alternating Stationary Distribution Correction Estimation (Matsunaga et, al. NeurIPS 2023)

[2] OptiDICE: Offline Policy Optimization via Stationary Distribution Correction Estimation (Lee et, al. ICML 2021)

[3] Offline Multi-Agent Reinforcement Learning with Implicit Global-to-Local Value Regularization (Wang et, al. NeurIPS 2023)

----------------------------------------------------------------------------------------------------------------------
Post-Rebuttal
----------------------------------------------------------------------------------------------------------------------

During the rebuttal, the authors worked hard to address my concerns, namely (1) additional baselines (AlberDICE and OptiDICE) on all 3 environments, (2) additional toy example (XOR Game), (3) a re-interpretation of the method as learning the optimal policy implicitly in factorized policy space and (4) detailed analysis on how ComaDICE works based on the XOR Game which is a toy domain.

The reason why my initial score was low was because a lot of these points were not explicit in the original draft, and it was unclear what the contribution was. After the rebuttal and some discussion, these points were made clear and the authors demonstrated that (1) using a factorized approach to decomposing stationary distributions is implicitly searching for optimal factorized policies (2) ComaDICE is able to approximate a global optimum as opposed to Nash policies (AlberDICE), (3) ComaDICE outperforms all baselines in all environments including AlberDICE and OptiDICE, which are most closely related, (4) the algorithm is more scalable and simple compared to AlberDICE.

As it currently stands, these contributions of the paper are scattered and it would take some effort by the reader to appreciate the paper, without some background in both DICE-based approaches in MARL. I would recommend a re-write of the storyline of the paper, highlighting the fact that ComaDICE pushes the cooperative Offline MARL field forward beyond IGM as well as AlberDICE.

As a result, I've increased my score as follows:

Soundness: 2-->3

Presentation: 1-->2

Contribution: 1--> 3

Overall Score: 3: reject, not good enough --> 6: marginally above the acceptance threshold

I can't quite give an 8 (Strong Accept) due to the writing, but I would have given 7 if that option was available.

---

> ### Author Response · Authors · 2024-11-22
> **We thank the reviewers for the feedback!**
>
> We thank the reviewer for the comments. Please find below our responses to your concerns.
>
> > A very optimistic reading of the paper would interpret that the decomposition procedure is actually learning optimal value functions over decentralized policies.  ...  However, it is worth noting that this is more of a statement of the potential of the paper rather than a strength of its current version, as these points are not specifically addressed in the current draft.
> Furthermore, ComaDICE can be more scalable in comparison to AlberDICE [1] which requires alternating optimization.
>
> In this paper, we propose a method to learn a globally optimal state-action occupancy by integrating the DICE framework with decentralized learning principles. While the DICE framework provides an effective approach for addressing out-of-distribution (OOD) issues, as demonstrated in prior DICE-based offline RL studies, the decentralized learning principle ensures that our algorithm remains scalable and efficient. Compared to AlberDICE, in addition to being more scalable, as you mentioned, our algorithm is better in capturing the interdependencies between local agents and handling credit assignment across agents through the use of a mixing architecture.
>
> > It is not clear what the main problem is that ComaDICE is solving.  ....
>
> Our algorithm addresses the OOD issue in a manner similar to other DICE-based offline RL algorithms, such as OptDICE and AlberDICE. Specifically, alongside maximizing the global reward, we incorporate a divergence term into the objective function to ensure the learned policy remains close to the behavior policy. The main distinction between our algorithm and AlberDICE lies in the learning approach: our method learns the occupancy ratio at the global level, leveraging centralized learning and decentralized execution. In contrast, AlberDICE focuses on learning individual occupancy ratios, which limits its ability to capture the interconnections between local agents. We have added clarifying details on Page 3 to highlight these differences.
>
> > The derivation is based on OptiDICE [2] but this is not explicitly mentioned,  ...  Furthermore, the final algorithm closely resembles OMIGA [3].
>
> We would like to note that while OptiDICE is designed for the single-agent setting, our work extends its DICE-based approach to the multi-agent setting, which requires an extensive analysis on the connection between local policies and the global objective. Theoretically, we have included a closed-form expression for the first-order derivative of the objective function in our Proposition 4.1, which we believe is not available in the OptiDICE paper. To clarify this, we have added a detailed explanation after our proposition on Page 5.
>
> Additionally, we have included discussions on Page 3 to highlight the differences between our algorithm, AlberDICE, and OMIGA. Furthermore, new numerical experiments have been provided to compare our algorithm with AlberDICE and OptiDICE.
>
> > What is the main purpose for the value decomposition and learning individual Q-functions?
>
> The primary motivation for decomposing the Q-function is to enable the decomposition of the advantage function, which is a key component of our objective. In our multi-agent setting, local rewards are unavailable, making it challenging to directly compute local advantage functions. To address this, we learn local Q-functions and use them to derive the local advantage functions. Simultaneously, the mean squared error (MSE) is optimized to ensure that the global Q-function and state-value function align well with the global rewards. An explanation has been added to Page 7 to clarify this point.
>
> > AlberDICE and OptiDICE are missing as the main baselines.
>
> We have included OptiDICE  and AlberDICE  for comparison and updated our experiments.
>
> >What is the purpose of Theorem 4.2, ..
>
> Our theorem demonstrates that the loss function is concave with respect to νν when any mixing network with non-negative weights and convex activations is employed. This guarantees that the training process will remain stable when optimizing over $\nu$. Consequently, under our mixing architecture, minimizing the loss with respect to $\nu$ ensures that the global optimum will be achieved.

---

> ### Author Response · Authors · 2024-11-22
>
> > Can ComaDICE solve the XOR game and Bridge mentioned in AlberDICE?
>
> Yes, ComaDICE is capable. In fact, these games are relatively small, with significantly lower state and action dimensions, allowing our algorithm to quickly converge to the optimal policy within just a few training epochs. For the XOR game, we observed that ComaDICE not only achieves the maximum possible reward of 100 but also converges much faster than AlberDICE. We have added such results to Section B.7. in appendix.
>
> > Related to “Strengths”, can solving ComaDICE learn the global optimum in the underlying MDP even with the mixing network?
>
> Yes, under our mixing architecture, ComaDICE can theoretically learn the global optimum for $\nu_i$ (for all $i$), as guaranteed by the convexity with respect to $\nu$ established in Theorem 4.2.
>
> > Is the Individual Global Max (IGM) assumption required in order to introduce the mixing network? In other words, does ComaDICE assume that the underlying optimal Q function assume IGM?
>
> We do not assume IGM in our approach. Instead, we demonstrate that our learning framework guarantees consistency between the global and local optimal policies, as shown in Proposition 4.3—a property that shares similarities with IGM. Our mixing network architecture is primarily designed to ensure the convexity of the loss function, enabling a stable and efficient training process.
>
> > What is the main difference in the final algorithm with OMIGA [3] ?
>
> The fundamental difference is that OMIGA directly produces Q or V functions, which can then be used to extract a policy. In contrast, ComaDICE uses the Q and \(\nu\) functions solely to learn the occupancy ratio between the learning policy and the behavioral policy. We have added a discussion on Page 3 to clarify this distinction.
>  > Why are the notations mixed between using joint observations  and global states
>
> In the POMDP setting, global states are not fully accessible and are instead represented by the joint observations from local agents. For simplicity, we use global state notation, but it actually refers to the corresponding joint observations. We have added a clarification on Page 4 at the beginning of Section 4.1.
>
> > Please write in a different color during the rebuttal which part is the novel part. In particular, which part of the proofs are a novel contribution and which part is coming from OptiDICE [2]
>
> Thank you for the suggestion. We have added some lines to the proof of our Proposition 4.1, acknowledging that the first part of the proof is a straightforward extension from the OptDICE paper, while the second part introduces some novel findings.
>
> **We hope our revisions effectively respond to the reviewer’s critiques and provide greater clarity on our contributions. If you have any additional questions or comments, we would be happy to address them.**

---

> ### Comment · Reviewer_kymz · 2024-11-23
> **I appreciate the effort by the authors. But some remaining concerns..**
>
> I'd like to thank all of the authors for the hard work during the rebuttal, especially in providing extensive experiments with additional baselines and extra toy example (XOR Game) results. It seems pretty clear that ComaDICE performs well across a variety of environments and outperforms baselines, including AlberDICE and OptiDICE. I believe these new results strengthen the paper.
>
>
> However, I still have some remaining concerns and questions so I hope you can clarify them:
>
> > **Compared to AlberDICE, in addition to being more scalable, as you mentioned, our algorithm is better in capturing the interdependencies between local agents and handling credit assignment across agents through the use of a mixing architecture.**
>
> > **In contrast, AlberDICE focuses on learning individual occupancy ratios, which limits its ability to capture the interconnections between local agents.**
>
> While AlberDICE does learn individual occupancy ratios, they are solving a reduced MDP which also considers the other agents' current policy and their training procedure also falls under CTDE.
>
> In detail, what are the core differences which allow ComaDICE to learn interconnections between local agents and handling credit assignment? It would be helpful if you can write a detailed comparison using both formulations.
>
> Also on a related note, I would suggest emphasizing the ability for ComaDICE to learn a factorized policy which is a global optimum. This would also clarify the strength of ComaDICE over AlberDICE (which learns Nash policies).
>
> >**Yes, under our mixing architecture, ComaDICE can theoretically learn the global optimum for $\nu_i$ (for all $i$), as guaranteed by the convexity with respect to
>  established in Theorem 4.2.**
>
> However, I still have concerns whether it does indeed a globally optimum $\nu$. Theorem 4.2 is used to claim that $ \mathcal{\tilde L}$ is convex in $\nu$, but does solving this lead to the same $\nu$ as in the original problem without value decomposition? My understanding is that this is not the case, and either an assumption (e.g. IGM) or a separate theorem is required.
>
> Finally, regarding notation, I think it would be much simpler if states are used throughout the paper rather than joint observations. The main reason is that, as far as I understand, the derivations and the proof actually require the global state rather than joint observations (which are not equivalent to the state). Of course, partial observability can be introduced in the practical algorithm in Section 5.

---

> > ### Author Response · Authors · 2024-11-23
> > **We thank the reviewer for the additional feedback!**
> >
> > We thank the reviewer for their prompt and valuable feedback, which we greatly appreciate. We also thank the reviewer for highlighting OptDICE and AlberDICE during the rebuttal process, as this has significantly improved our paper and helped us better clarify our contributions.
> >
> > > In detail, what are the core differences which allow ComaDICE to learn interconnections between local agents and handling credit assignment? It would be helpful if you can write a detailed comparison using both formulations.
> >
> > We thank the reviewer for the question. We believe the key difference that enables ComaDICE to better capture the interconnections between agents lies in its approach to learning global values and its factorization method, which integrates information from local agents to construct the global value and global occupancy functions. Specifically, we propose learning a global occupancy ratio and leveraging a factorization approach to decompose the global learning variables into local ones, using local information. This approach will help capture how each local agent's contribute to the global objective. This is also the common advantage of such value factorization approaches.
> >
> > Additionally, in our policy extraction phase, we learn local policies using a shared global occupancy ratio, $w^{tot}$ (Eq. 9 in our paper). We believe this design inherently addresses aspects of credit assignment across agents, which is a feature lacking in AlberDICE.
> >
> > > Also on a related note, I would suggest emphasizing the ability for ComaDICE to learn a factorized policy which is a global optimum. This would also clarify the strength of ComaDICE over AlberDICE (which learns Nash policies).
> >
> > We thank the reviewer for the suggestion. This is indeed a key advantage of ComaDICE over AlberDICE, and we will make sure to emphasize this point in the revised manuscript.
> >
> > > However, I still have concerns whether it does indeed a globally optimum $\nu$. Theorem 4.2 is used to claim that $ \mathcal{\tilde L}$ is convex in $\nu$, but does solving this lead to the same $\nu$ as in the original problem without value decomposition? My understanding is that this is not the case, and either an assumption (e.g. IGM) or a separate theorem is required.
> >
> > We thank the reviewer for the insightful comment. We agree that the mixing network approach imposes restrictions on the space of $\nu$, which could result in the solution to $\min \mathcal{\tilde L}$ being suboptimal and potentially different from the one derived from the original objective (without decomposition). However, it is well-established that a two-layer feedforward neural network with non-linear activations, such as ReLU, possesses **universal approximation** capabilities. Therefore, in theory, the mixing network can approximate any global $\nu$ value, implying that solving $\min \mathcal{\tilde L}$ could return the global optimum for the original problem.
> >
> > As an additional note, if the mixing network is a simple linear combination (as used in our experiments), the solution to $\min \mathcal{\tilde L}$ may indeed be suboptimal unless additional assumptions are satisfied, such as the global $\nu$ being linearly decomposable. We will incorporate this discussion into the updated paper to clarify this point.
> >
> > >Finally, regarding notation, I think it would be much simpler if states are used throughout the paper rather than joint observations. The main reason is that, as far as I understand, the derivations and the proof actually require the global state rather than joint observations (which are not equivalent to the state). Of course, partial observability can be introduced in the practical algorithm in Section 5.
> >
> > We thank the reviewer for the suggestion, which we find very useful. While we actually prefer this approach, we previously encountered feedback suggesting that global states are not fully available and are instead represented by joint observations, thus using global states would be confusing. This led us to use a mixture of global states and joint observations in our formulation. We are happy to adopt the notation you suggested and will update the paper accordingly.
> >
> > **We hope our responses address your concerns. We also sincerely thank the reviewer for their additional suggestions, which are very valuable for further improving the paper. If you have any further questions or comments, we would be happy to address them.**

---

> > > ### Comment · Reviewer_kymz · 2024-11-24
> > >
> > > Thank you for the clarifications.
> > >
> > > Can you confirm whether the following statement in my original review is true?
> > >
> > > > It is known that every MDP has a deterministic optimal policy, which, if extended to the multi-agent setting, can be factorized into independent policies. A very optimistic reading of the paper would interpret that the decomposition procedure is actually learning optimal value functions over decentralized policies.
> > >
> > > If we view the goal of MARL as $\max_{\pi_{tot}}J(\pi_{tot})$, where $\pi_{tot}$ is the space of factorized policies, is this what the objective $\mathcal{\tilde L}$ (Line 262) is doing? I am having trouble understanding clearly how ComaDICE is able to do this, and solve the XOR Game for example. It would also be helpful if you can provide what the converged $\nu, q, A_\nu $ and $w$ values (both global and individual values) were for the XOR Game for better understanding.

---

> ### Author Response · Authors · 2024-11-24
> **We thank the reviewer for the additional feedback!**
>
> We thank the reviewer for the prompt additional feedback, which we highly appreciate.
>
> > Can you confirm whether the following statement in my original review is true? ...
>
> We believe your observations align well with the high-level perspective of our method. To clarify further, our approach involves learning a global policy in the form of an occupancy measure by decomposing the Lagrange multiplier in the DICE framework, i.e., $\nu$, which can be interpreted as a value function. Subsequently, our policy extraction step focuses on learning decentralized policies derived from the outcome of the DICE approach, specifically $w^{tot}$.
>
> > If we view the goal of MARL as $\max L(\pi^{tot})$ , where  is the space of factorized policies, is this what the objective of $\max \widetilde{L}$ (Line 262) is doing?
>
> Thank you for the question. To clarify, the objective of $\min \widetilde{L}$ is also to find an optimal $\pi^{tot}$, but it does so within the space of **occupancy measures**. Here, instead of factorizing the global policy directly, we factorize the Lagrange multiplier $\nu$. As a result, $\min \widetilde{L}$ does not output a globally optimal policy directly, but rather the *occupancy ratio* between the optimal policy and the behavior policy. The optimal policy is then extracted from this occupancy ratio through weighted behavior cloning (BC), where we propose learning *decentralized policies* that match with the resulting *occupancy ratio.*
>
> > I am having trouble understanding clearly how ComaDICE is able to do this, and solve the XOR Game for example. It would also be helpful if you can provide what the converged  and  values (both global and individual values) were for the XOR Game for better understanding.
>
> We thank the reviewer for the question. We believe that, for the XOR game, our algorithm operates in a manner similar to AlberDICE. Both approaches aim to learn policies that maximize the reward while aligning with the behavioral policies. However, whereas AlberDICE performs this alignment at the level of individual agents, our approach operates at the global level.
>
> For the XOR game, we have all numerical results demonstrating how ComaDICE achieves the optimal values reported in the paper. However, let us explain intuitively how ComaDICE solves the XOR game. Consider the dataset {AB}, where this observation yields a high reward (i.e., 1). When ComaDICE solves $\min \mathcal{L}$, it seeks a policy that maximizes the reward across the dataset while aligning with the behavioral policy represented by the dataset {AB}. Consequently, it will return a global optimal policy (in the form of an occupancy ratio) that assigns the highest possible probabilities to the joint actions {AB}.
> Subsequently, our weighted behavior cloning (BC) step learns decentralized policies that also assign the highest possible probabilities to the joint actions {AB}, returning the desired optimal policy observed in our experiments. The same reasoning applies to other datasets, such as {AA, AB, BA}, ensuring that ComaDICE learns the correct optimal policies across different scenarios.
>
> **We hope our responses address your concerns. If the reviewer would like a more detailed explanation of the policy values returned by our algorithm for the XOR game, we would be happy to provide additional clarifications and discussions.**

---

> ### Comment · Reviewer_kymz · 2024-11-25
>
> Regarding whether $\mathcal{\tilde L}$ is learning the optimal policy over factorized policies, I understand the optimization is over occupancy measures. I am asking whether the **implicitly** learned optimal policy is over factorized policies (before policy extraction). For instance, perhaps it is possible to say that since $\rho (s, a) = \rho(s) \pi(a|s)$ where $\rho(s)$ is the state marginal, and we want $\pi(a|s)$ to have a factorized form, factorizing the value is actually learning an optimal stationary distribution over factorized policies. This is not exact and just a general idea but something similar would help understand intuitively what value factorization is doing here.
>
> Regarding the Matrix Game, can you explain intuitively how ComaDICE is able to solve the XOR Game using the ${AB, BA}$ dataset? For instance looking at Table 26, it seems like $w(AB) $ would converge to some positive value while $w(BA) = 0$ in order to deterministically choose AB. How  is ComaDICE able to do this despite both AB and BA being in the dataset?

---

> > ### Author Response · Authors · 2024-11-25
> >
> > > Regarding whether $\widetilde{L}$ is learning the optimal policy over factorized policies ...
> >
> > We thank the reviewer for the additional question, which provides an opportunity to further discuss and clarify our contributions. Our learning objective operates in the space of occupancy measures, and it is generally challenging to prove that optimizing this objective is equivalent to optimizing over factorized policies, particularly when incorporating the f-divergence and a non-linear mixing network. However, under certain conditions, we believe that such factorization properties can be observed.
> >
> > Specifically, when examining the objective function $\widetilde{L}$ on Page 3, if the mixing network has a linear structure and the f-divergence is chi-square (as employed in our experiments), the inverse function $f^{-1}$ also takes a linear form. In this case, the global objective function can be decomposed into local objective functions with variables $\nu_i$. Minimizing $\widetilde{L}$ under these conditions is approximately equivalent to optimizing each local objective function $\widetilde{L}_i$. Minimizing these local functions yields local occupancy ratios $\rho_i / \mu_i$.
> >
> > Thus, under this linearity setting, optimizing the global function $\widetilde{L}$ effectively approximates learning factorized occupancy ratios.
> >
> > In addition, under a more general setting, proving the equivalence under a general f-divergence function and a two-layer mixing network is challenging. However, if we consider linear combinations that approximate the non-linear f-divergence and mixing network, we can assert that optimizing our global function is approximately equivalent to learning factorized occupancies.
> > We hope this explanation clarifies how our value factorization method operates. We will incorporate this discussion into the updated version of the paper.
> >
> > > Regarding the Matrix Game, can you explain intuitively how ComaDICE is able to solve the XOR Game using the {AB,BA} dataset.
> >
> > We thank the reviewer for the question. Both $\pi(AB) = 1$ (and the others =0) or $\pi(BA) = 1$ are optimal solutions to our learning problem. In our experiments, ComaDICE converges to the case where $\pi(AB) = 1$ and $\pi(BA) = 0$.
> >
> > Let us explain why this happens. Our objective function consists of two terms: one aims to maximize the reward, and the other minimizes the divergence between the learned policy and the dataset. When the dataset consists of $\{AB, BA\}$, the occupancy-matching term tends to favor a policy that assigns positive probabilities to both $AB$ and $BA$. However, if both $AB$ and $BA$ have (significantly) positive probabilities, this implies that the first player would take both actions A and B with some positive probability, leading to a lower reward. In other words, exactly matching the dataset distribution would result in low reward.
> >
> > Thus, to optimize the overal objective, ComaDICE assigns the highest probability to one action pair (in this case, $AB$), ensuring that the policy achieves a better balance between maximizing the reward and minimizing divergence. This explains why ComaDICE converges to this solution.
> >
> >
> > **We hope our responses address your concerns. If the reviewer has additional questions or wishes to continue this discussion further, we would be happy to provide further clarifications and engage in additional discussions**

---

> > > ### Comment · Reviewer_kymz · 2024-11-26
> > >
> > > Thank you for the answers. My concern regarding the first question about $\mathcal{\tilde L}$ is resolved and I think this kind of analysis would help clarify the contributions of ComaDICE.
> > >
> > > For the XOR Game, I think that only partially explains how ComaDICE can solve it because balancing reward maximization and conservatism would also apply to OptiDICE (which fails). However, the answer to the first question regarding $\mathcal{\tilde L}$ somewhat answers this question (reward maximization + conservatism in the factorized policy space).
> > >
> > > I think the paper can be significantly improved if the XOR (or other toy example) is used to illustrate the purpose of decomposition. It would be better to provide a characterization of $\nu$ in the toy example and show that decomposing it in that manner is implicitly finding optimal factorized policies.
> > >
> > > I am willing to raise my score if these points (along with all discussions during the rebuttal) are incorporated into the draft before the revision deadline.

---

> > > > ### Author Response · Authors · 2024-11-26
> > > >
> > > > We thank the reviewer for continuously engaging in the discussion and providing additional feedback. We have now incorporated our discussions into the updated version of the paper. Specifically, we have added **Section B1** to the appendix to detail our discussion on the factorization aspect of ComaDICE. Additionally, we have expanded **Section B8** of the appendix to explain how ComaDICE solves the XOR game. In particular, we have included discussions on how ComaDICE provides the correct optimal policies for the datasets under consideration and why OptDICE fails in these examples. All updates are highlighted in magenta for easy reference.
> > > >
> > > > We hope these updates address your remaining concerns. If you have further questions or suggestions, we would be happy to address them and make additional updates to the paper before the revision deadline.

---

> > > > > ### Comment · Reviewer_kymz · 2024-11-27
> > > > >
> > > > > Thank you again for the rebuttal and the discussion. I've updated my score which reflects my current view on the paper, as well as some details on the reasoning.

---

> > > > > > ### Author Response · Authors · 2024-11-27
> > > > > >
> > > > > > We sincerely thank the reviewer for all the valuable discussions and insightful questions during the rebuttal process. Your suggestions have pushed us to improve our paper significantly with additional experiments and more in-depth discussions. *We believe this exemplifies the kind of constructive and thoughtful review that any author would hope to receive.*
> > > > > >
> > > > > > We will ensure additional rounds of revision to refine the storyline and enhance the overall writing quality of the paper.

---

### Official Review · Reviewer_RYfn · 2024-11-03

**Soundness:** 3
**Presentation:** 2
**Contribution:** 3
**Rating:** 6
**Confidence:** 3

**Summary:**

This paper proposes an algorithm introducing stationary distribution correction to address the distributional shift problem in offline cooperative multi-agent reinforcement learning (MARL). In multi-agent environments, this issue is intensified due to the large joint state-action space and the interdependencies among agents. To tackle this, ComaDICE minimizes the f-divergence between the stationary distributions of the learning and behavior policies. Additionally, by leveraging the Centralized Training with Decentralized Execution (CTDE) framework, it decomposes the global value functions into local values for each agent, ensuring that the optimization of each agent’s local policy is consistent with the global learning objective.

**Strengths:**

- Effective Distribution Correction in Multi-Agent Settings: ComaDICE improves upon traditional DICE by extending stationary distribution correction to multi-agent environments. Through f-divergence-based alignment between behavior and target policies, it effectively handles the complex distributional shifts unique to multi-agent interactions, enhancing policy reliability and performance.
- Theoretical Foundation: A thorough mathematical analysis provides convergence and stability proofs through f-divergence correction, reinforcing the algorithm’s reliability in multi-agent scenarios.
- Stable Value Decomposition: ComaDICE decomposes global values into convex local objectives, enhancing training stability and aligning local agent optimization with the global objective to address coordination and stability challenges specific to multi-agent environments.

**Weaknesses:**

- ComaDICE’s theoretical analysis relies on non-negative weights and convex activations in the mixing network for stability. While this aids convergence, it may limit the model’s ability to capture complex inter-agent dynamics. In the practical algorithm, this limitation is intensified by the use of a single-layer linear mixing network, which further restricts representational capacity in highly interactive environments.
- High Computational Cost: ComaDICE incurs a significant computational cost due to the precise f-divergence-based adjustments needed for each agent’s local policy in stationary distribution correction. This can lead to reduced efficiency, especially in environments with a large number of agents.

**Questions:**

The ablation study shows that performance decreases when using a more complex network. Why is that the case? If more data were available or the model was improved, could the results potentially differ?

---

> ### Author Response · Authors · 2024-11-22
> **We thank the reviewers for the feedback!**
>
> We thank the reviewer for the comments. Please find below our responses to your concerns.
> > ComaDICE’s theoretical analysis relies on non-negative weights and convex activations in the mixing network for stability. While this aids convergence, it may limit the model’s ability to capture complex inter-agent dynamics...
>
> We would like to emphasize that non-negative weights and convex activations have been widely used in prior MARL algorithms with mixing architectures, such as QMIX, QTRAN. Our choice to adopt this setting is driven by its crucial role for achieving strong algorithmic performance that was experimentally demonstrated in previous studies. For instance, [1] shows that mixing networks with negative weights or non-convex activations result in significantly worse performance. We added some lines on Page 6 to clarify this point.
>
> [1] The Viet Bui, Tien Mai, and Thanh Hong Nguyen. Inverse factorized q-learning for cooperative multi-agent imitation learning. Advances in Neural Information Processing Systems, 38, 2024.
>
> > High Computational Cost: ComaDICE incurs a significant computational cost ...
>
> In our algorithm, the use of f-divergence is particularly advantageous. Its closed-form formulation and convexity enable a closed-form solution for the inner maximization problem, allowing us to reformulate the min-max problem into a non-adversarial one. This significantly enhances the training process. Importantly, our experiments clearly demonstrate that the algorithm is highly scalable and efficient, even for SMACv2, which is widely regarded as one of the most high-dimensional and challenging MARL benchmarks.
>
> > The ablation study shows that performance decreases when using a more complex network. Why is that the case? If more data were available or the model was improved, could the results potentially differ?
>
> The low performance of the 2-layer mixing network shows that, in our offline setting, the 2-layer structure is overly complex for capturing the interdependencies between agents, leading to overfitting. Increasing the amount of offline data might reveal more information and potentially improve the performance of the 2-layer mixing network. However, this is not practical, as it would require significantly more storage and make training computationally expensive and infeasible.
>
> We have added an additional discussion to Section B.6 in the appendix to clarify this point. If our explanation remains unclear or if you have any further questions, we would be happy to provide additional clarification.
>
> **We hope our revisions address the reviewers’ critiques and further clarify our contributions. If there are any additional questions or comments, we would be happy to address them.**

---

> > ### Comment · Reviewer_RYfn · 2024-11-24
> >
> > Thank you for your detailed response. I appreciate the effort to address my concerns. But when I consider the contribution of the paper, I'd like to maintain my score.

---

> > > ### Author Response · Authors · 2024-11-24
> > >
> > > We thank the reviewer for taking the time to read our responses and provide additional feedback.

---

### Author Response · Authors · 2024-11-22
**We thank the reviewers for their feedback!**

We sincerely thank the reviewers for their constructive and thoughtful comments, which we have addressed to the best of our ability. Our response was slightly delayed as we worked to incorporate additional baselines into our benchmarking environments, which required significant time. One reason for the delay was that the published source code for the AlberDICE paper did not support SMAC environments, necessitating communication with the authors to obtain the required code, which took additional time. We now have a complete set of updated results and are ready to respond to your feedback.

To address the reviewers’ concerns, we have made the following major updates to the paper:
- Included two DICE-based MARL algorithms for comparison (OptDICE and AlberDICE).
- Added experiments on the XOR game environment (as used in the AlberDICE paper).
- Clarified several points raised in the feedback.

All updates are highlighted in magenta. We hope our revisions address the reviewers’ critiques and further clarify our contributions. If there are any additional questions or comments, we would be happy to address them.

---

### Meta-Review · Area_Chair_nZ1L · 2024-12-19

**Metareview:**

The paper studied offline multi-agent RL and proposed an approach based on the DICE framework. The proposed approach uses a stationary distribution shift regularization to combat the distribution shift issue in offline RL. The paper demonstrates that their approach works well empirically.

**Additional Comments On Reviewer Discussion:**

The reviewers are in general positive about the paper. Before rebuttal, the reviewers had concerns about the experiments (additional baselines and environments), and comparison to DICE based related works. During the rebuttal,  the authors worked hard to provide additional details, including additional baselines (AlberDICE and OptiDICE) on all 3 environments, additional toy example (XOR Game), and a re-interpretation of the method as learning the optimal policy implicitly in factorized policy space, and detailed analysis on how ComaDICE works based on the XOR Game. These additions convinced one reviewer to increase their score to the positive side, and at the end all reviewers agreed on an acceptance.

---

### Decision · Program_Chairs · 2025-01-22

Accept (Poster)